# Chimeric antigen receptor T cells targeting FcRH5 provide robust tumour-specific responses in murine xenograft models of multiple myeloma

Dongpeng Jiang [1,3], Haiwen Huang[2,3], Huimin Qin[1,3], Koukou Tang [1], Xiangru Shi [1], Tingting Zhu [1], Yuqing Gao [1], Ying Zhang[1], Xiaopeng Tian [2], Jianhong Fu [2], Weiwei Qu [1], Weilan Cai [1], Yang Xu [1,4] ✉, Depei Wu [1,4] ✉ & Jianhong Chu [1,4] ✉

BCMA-targeting chimeric antigen receptor (CAR) T cell therapy demonstrates impressive clinical response in multiple myeloma (MM). However, some patients with BCMA-deficient tumours cannot benefit from this therapy, and others can experience BCMA antigen loss leading to relapse, thus necessitating the identification of additional CAR-T targets. Here, we show that FcRH5 is expressed on multiple myeloma cells and can be targeted with CAR-T cells. FcRH5 CAR-T cells elicited antigen-specific activation, cytokine secretion and cytotoxicity against MM cells. Moreover, FcRH5 CAR-T cells exhibited robust tumoricidal efficacy in murine xenograft models, including one deficient in BCMA expression. We also show that different forms of soluble FcRH5 can interfere with the efficacy of FcRH5 CAR-T cells. Lastly, FcRH5/BCMA-bispecific CAR-T cells efficiently recognized MM cells expressing FcRH5 and/or BCMA and displayed improved efficacy, compared with mono-specific CAR-T cells in vivo. These findings suggest that targeting FcRH5 with CAR-T cells may represent a promising therapeutic avenue for MM.

Chimeric antigen receptor (CAR) T cell therapy has emerged as a very promising immunotherapeutic approach for relapsed and refractory multiple myeloma (RRMM). B cell maturation antigen (BCMA) is generally regarded as a suitable target antigen by virtue of its restricted expression on normal and malignant plasma cells a well as some subsets of B cells[1]. Despite 64–85%[2–4] overall response rates following initial BCMA CAR-T treatment, relapses were frequently observed in approximately 45% of responders[5]. Downregulation or loss of BCMA is one of the potential mechanisms though which patients relapse and

occurs in 4–33% of progressive patients following BCMA CAR-T cell therapy[6,7]. Two recent studies have identified homozygous deletion of BCMA or biallelic loss of BCMA as the underlying mechanisms for immune escape following BCMA CAR-T therapy[8,9]. Another potential disadvantage for BCMA as a CAR-T target lies in that BCMA can be actively shed from the MM cell surface by γ-secretase complex, which in turn lowers surface BCMA expression on MM cells for CAR-T cell recognition and also releases the soluble form of BCMA, which can diminish CAR T cell effector function[10]. In addition, BCMA has been

[1]Institute of Blood and Marrow Transplantation, National Clinical Research Center for Hematologic Diseases, Jiangsu Institute of Hematology, The First Affiliated Hospital of Soochow University, Collaborative Innovation Center of Hematology, Soochow University, Suzhou, Jiangsu, China. [2]Department of hematology, The First Affiliated Hospital of Soochow University, Suzhou, Jiangsu, China. [3]These authors contributed equally: Dongpeng Jiang, Haiwen Huang, Huimin Qin. [4]These authors jointly supervised this work: Yang Xu, Depei Wu, Jianhong Chu. ✉e-mail: yangxu@suda.edu.cn; wudepei@suda.edu.cn; chujianhong@hotmail.com

reported to be primarily absent or dimly expressed on MM cells from some patients, which may preclude treatment with BCMA CAR-T cells[2,11,12]. On the other hand, we and others have previously identified CS1 as a candidate target antigen for CAR-T cells in MM in the pre-clinical studies[13,14], and also several CS1 CAR-T cell products have entered the pipeline[15]; however, the safety concern about the potential on-target off-tumor toxicity of CS1 CAR-T cells has been raised since CS1 is also present on subsets of normal lymphocytes including T cells, natural killer (NK) cells and NKT cells. Therefore, exploring alternative targetable antigens by CAR-T cells with favorable safety profile to mitigate antigen loss and cure MM patients with dim BCMA expression remains urgently desirable.

Fc receptor-homolog 5 (FcRH5, also known as CD307, IRTA2) is a differentiation antigen homologous to the family of Fc receptors, which is exclusively expressed in the B cell linage. FcRH5 can be detected as early as in pre-B cells, and its expression is retained on normal plasma cells, unlike other B cell surface markers, such as CD19 and CD20[16]. FcRH5 expression is upregulated on malignant plasma cells in MM in comparison to normal plasma cells[17,18], especially on those with ampli-fication or gain of chromosome 1q21, representative of a very high-risk genetic feather[19]. Moreover, expression of FcRH5 was maintained on relapsed or refractory MM patients previously treated with proteasome inhibitor or immunomodulatory agents[20]. Immunotherapeutic approa-ches targeting FcRH5 have been developed in the attempt to cure MM. For example, the anti-FcRH5 T cell-dependent bispecific antibody (TDB) displayed potent anti-myeloma efficacy both in vitro and in a murine xenograft model[18]. In addition, the antibody-drug conjugate (ADC) DFRF4539A targeting FcRH5 has demonstrated appreciable anti-MM efficacy in xenograft models[17], but unexpectedly, showed limited clin-ical benefit in the phase-I trial recruiting RRMM patients[20].

In this study, we hypothesized that CAR-T cells targeting FcRH5 might provide more durable and robust anti-tumor efficacy compared with antibodies or ADCs with shorter half-life or potential immuno-genicity. We first confirmed that FcRH5 is expressed in MM. We demonstrated that FcRH5 CAR-T cells displayed specific and potent anti-MM activity both in vitro and in murine xenograft models, including a BCMA antigen loss model. Furthermore, we characterized the effect of two soluble forms of FcRH5 protein on the effector function of FcRH5 CAR-T cells. FcRH5 CAR-T cells showed minimal reactivity with normal B cells, and their safety could be further improved by incorporation of inducible caspase-9 'suicide' system. Lastly, bispecific CAR-T cells targeting FcRH5 and BCMA demonstrated appreciable MM-specific responses in vitro and in a xenogeneic mouse model. These findings support FcRH5 as a candidate antigen to target with CAR-T cells for the treatment of MM.

## Results

### Expression of FcRH5 on patient-derived MM cells and MM cell lines

We analyzed the expression of FcRH5 and BCMA on the surface of CD138[+] primary myeloma cells from bone marrow biopsies of 28 patients with MM by flow cytometry. As illustrated in Fig. 1a, expres-sion of either FcRH5 or BCMA on primary myeloma cells was detect-able, albeit with considerable inter-patient variability, consistent with previous reports[1,17,18]. The percentage of FcRH5-positive myeloma cells ranged from 10.5 to 99.7%, among which 22 patients (78.57%) expres-sed FcRH5 on more than 50% of myeloma cells. In contrast, the per-centage of BCMA-positive myeloma cells ranged from 1.44 to 96.44%, with only 10 patients (35.71%) expressing BCMA on more than 50% of myeloma cells (Fig. 1b). The MFIr (mean fluorescence intensity ratio) ranged from 1.12 to 10.9 (median MFIr 3.75) for BCMA staining in the myeloma cells of these patients, and from 2.57 to 57.45 (median MFIr 11.05) for FcRH5 staining (Fig. 1c). Importantly, we identified some patients displaying high expression of FcRH5 but negligible expression of BCMA with one representative result shown in Fig. 1d. In addition,

FcRH5 expression on myeloma cells seemed to be independent of BCMA expression in those patients (Fig. 1e). Moreover, the FcRH5 gene is localized in the chromosomal breakpoint in 1q21[21], and we found that the expression levels of FcRH5 protein were significantly elevated in the patients with 1q21 gain as compared to those without 1q21 gain (Supplementary Fig. 1a, b).

Although FcRH5 was highly expressed in patient-derived primary myeloma cells as previously demonstrated by others[17,20], and also by us here, lack of its expression on the commonly-used MM cell lines has been noted[18]. Using flow cytometry, we found FcRH5 protein was detectable on NCI-H929 cells, but almost absent on MM.1 s (Fig. 1f). Transduction with FcRH5 expression construct rendered MM.1 s cells abundantly expressing surface FcRH5 (Fig. 1f). In addition, Mino, a mantel cell lymphoma cell line, was found to express significant amount of native FcRH5 similar to a majority of primary MM cells (Fig. 1f), and was therefore used as target cells in the subsequent functional study of CAR-T cells.

### FcRH5 CAR T cells specifically recognize and lyse FcRH5[+] MM cells

We generated an expression construct encoding a second-generation CAR comprised of the anti-FcRH5 scFv linked to a CD8 hinge, and a CD28 transmembrane domain in tandem with a CD28.CD3 signaling endodomain (Fig. 2a), then introduced the CAR into primary bulk T cells to produce FcRH5 CAR-T cells by lentiviral transduction. T cells transduced with empty vector (mock T cells) were used as a control. Protein L staining-based flow cytometric analysis showed that the CAR was detectable on FcRH5 CAR-T cells (Supplementary Fig. 2a), which was confirmed by immunoblotting analysis with anti-CD3ζ antibody detecting chimeric anti-FcRH5-scFv-CD28-CD3ζ fusion protein (Sup-plementary Fig. 2b).

We then performed a series of in vitro function assays. We first checked the activation status of CAR-T cells in response to target cells by detecting expression of two activation markers CD69 and CD107a, and found that the percentage of CD69[+]CD107a[+] cells was increased in FcRH5 CAR-T cells stimulated with FcRH5[+] H929, Mino or MM.1s-FcRH5 cells, but not in those stimulated with FcRH5[-] MM.1 s cells (Supple-mentary Fig. 3a). Next, we performed a flow cytometry-based cyto-toxicity assay and observed that FcRH5 CAR-T cells efficiently lysed NCI-H929, Mino, MM.1s-FcRH5 cells but not MM.1 s cells, while mock T cells failed to elicit discernible cytolytic activity against any target cells (Fig. 2b). Moreover, the mean fluorescence intensity (MFI) of granzyme B was elevated in FcRH5 CAR-T cells in response to FcRH5[+] target cells, and the cytolytic capacity of FcRH5 CAR-T cells was substantially blunted when the calcium ion chelator EGTA capable of hindering release of granzyme B was supplemented (Supplementary Fig. 3b, c), suggesting that the cytolytic effect of FcRH5 CAR-T cells was largely dependent on the elevated expression and release of granzyme B.

Compared with mock T cells, FcRH5 CAR-T cells released sig-nificantly larger amounts of IFN-γ, IL-2 and TNF-α into the supernatants in response to FcRH5[+] target cells. However, when co-cultured with FcRH5[-] MM.1 s cells, FcRH5 CAR-T cells only produced background levels of cytokines, analogously to mock T cells (Fig. 2c). By monitoring the proliferation of T cells with eFluor670 dye dilution, we noticed that FcRH5 CAR-T cells underwent apparent cell division only when co-cultured with FcRH5[+] cells (Fig. 2d). The antigen dependent response of FcRH5 CAR-T cells was further validated using 293T cells over-expressing FcRH5 as target cells (Supplementary Fig. 4a–d).

To further evaluate the efficiency of FcRH5 CAR-T cells in the more clinically relevant context, we sought to explore whether patient-derived T cells engineered to express FcRH5-specific CAR could effi-ciently recognize and eradicate autologous MM cells. After confirming that CD138[+] primary MM cells isolated from three newly diagnosed MM patients expressed surface FcRH5 protein (Fig. 2e), we co-incubated these myeloma cells with autologous FcRH5 CAR-T cells

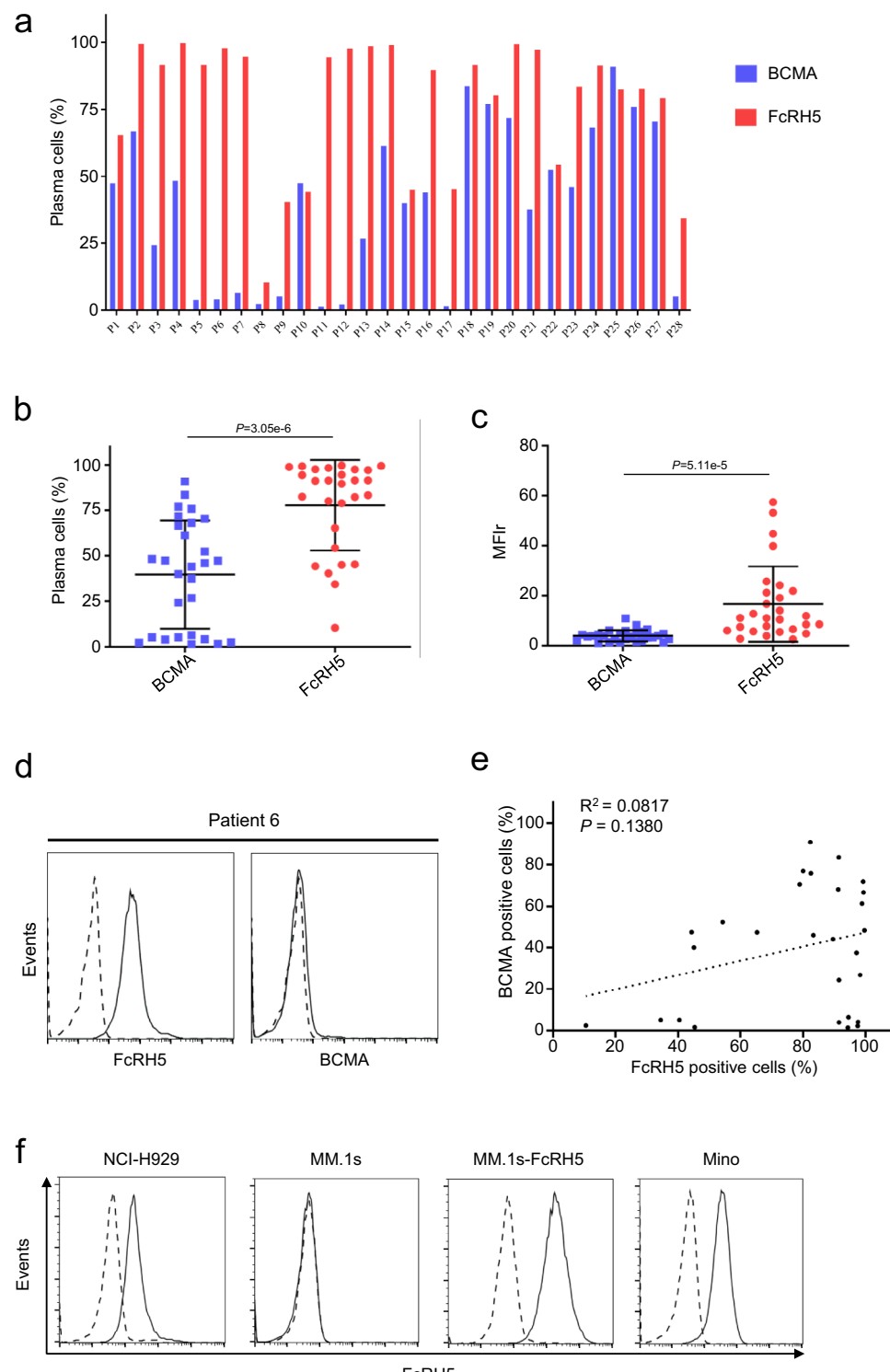

**Fig. 1 | Expression of FcRH5 and BCMA on primary myeloma cells from MM patients and MM cell lines. a** Expression of FcRH5 and BCMA protein in BMMC samples from 28 MM patients on the surface of CD138[+] myeloma cells as evaluated by Flow cytometry analysis. **b** Graph showed the mean of the percentage of BCMA-positive myeloma cells and FcRH5-positive myeloma cells in those patients. Mean ± SD, $n = 28$ patients, unpaired two-tailed $t$-test. **c** Graph depicted the mean of the MFIr (mean fluorescence intensity ratio) of BCMA and FcRH5 in the myeloma cells of those patients. Mean ± SD, $n = 28$ patients, unpaired two-tailed $t$-test. **d** Flow cytometric analysis of FcRH5 and BCMA expression on the surface of CD138[+] myeloma cells from the Patient 6. Staining with anti-FcRH5 mAb or anti-BCMA mAb (solid line) and isotype control (dashed line). **e** Correlation of FcRH5 expression with BCMA expression on the surface of CD138[+] myeloma cells as analyzed by linear regression analysis ($n = 28$). **f** Surface expression of FcRH5 on the indicated cell lines was measured by flow cytometry. FcRH5-overexpressing MM.1 s cells (MM.1s-FcRH5) was generated by lentiviral infection, and the lymphoma cell line Mino with abundant endogenous FcRH5 protein expression was used as a positive control.

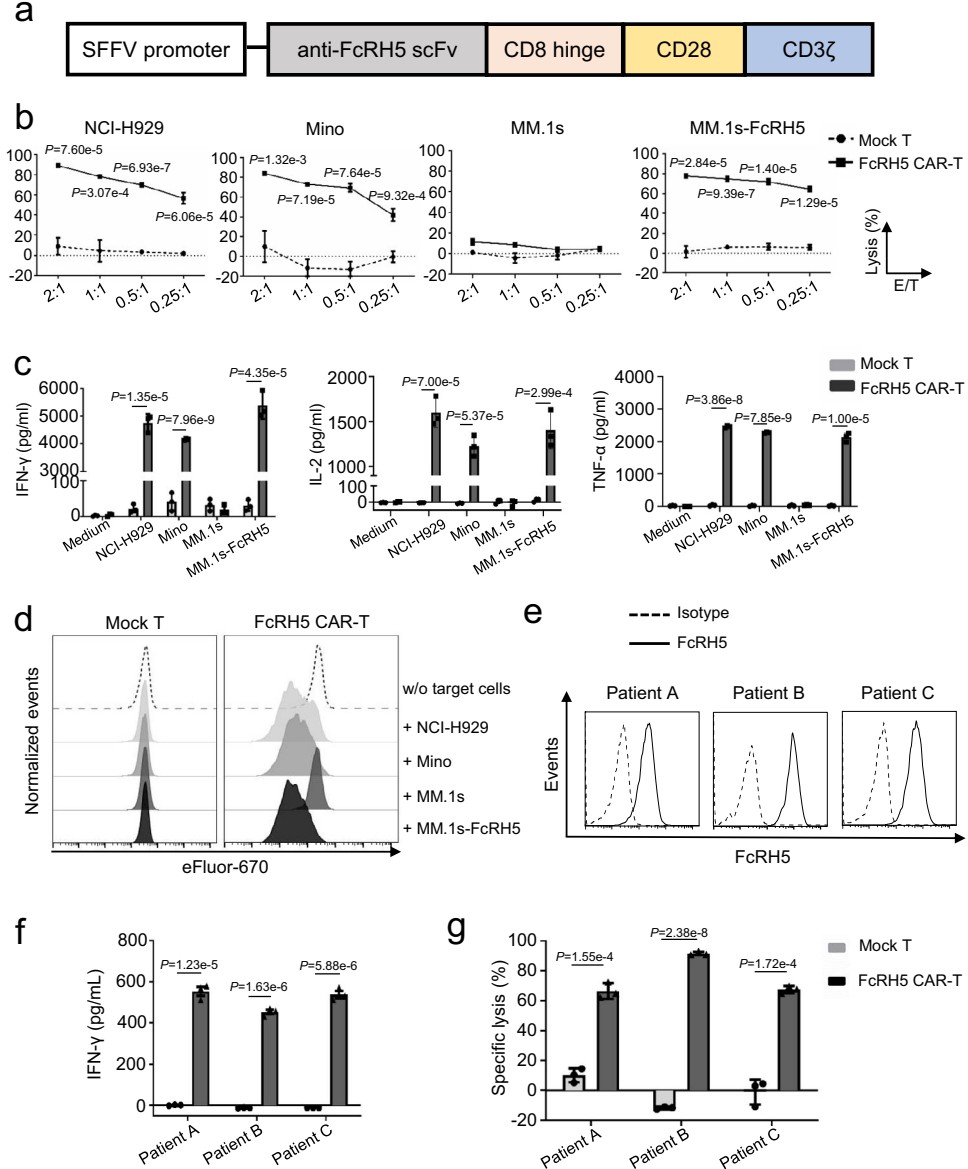

**Fig. 2 | FcRH5 CAR-T cells specifically recognize and lyse FcRH5⁺ target cells.**
**a** Schematic representation of FcRH5-specific CAR expression construct. **b** The cytolytic activity of mock T or FcRH5 CAR-T cells against indicated target cells at various effector: target cell (E/T) ratios was determined by a 6-h luciferase-based cytolytic assay. Data represent Mean ± SD from three independent experiments. **c** Mock T or FcRH5 CAR-T cells were co-cultured with target cells for 24 h, and cell-free supernatants were harvested for evaluating IFN-γ, IL-2 and TNF-α secretion. Mean ± SD, $n = 3$ independent co-cultures, unpaired two-tailed $t$-test. **d** Mock T or FcRH5 CAR-T cells were labeled with eFlour-670 and co-cultured with equal number of target cells for 5 days, and the dilution of eFlour-670 fluorescence signal was determined to reflect cell division. The plots were gated on CD3⁺ lymphocytes. The above experiments a-d were repeated with 2 different T cell donors. **e** Flow cytometric analysis of FcRH5 expression on CD138⁺ primary myeloma cells from three newly-diagnosed MM patients. Solid line represents staining with anti-FcRH5 mAb and dashed line represents staining with isotype control antibody. **f** The patient-derived myeloma cells were co-cultured with autologous mock T or FcRH5 CAR-T cells for 24 h, and cell-free supernatants were harvested for determination of IFN-γ secretion. Mean ± SD, $n = 3$ independent co-cultures, unpaired two-tailed Student $t$-test. **g** The patient-derived myeloma cells were labeled with eFlour-670 and co-cultured with autologous mock T or FcRH5 CAR-T cells at the E/T ratio of 5:1 for 6 h, and then cytolytic effect of T cells was determined by a flow cytometry-based assay. Mean ± SD, $n = 3$ independent co-cultures, two-tailed Student $t$ test.

for functional assays. As expected, FcRH5 CAR-T cells launched significantly augmented production of IFN-γ and 6-h cytolysis towards patient-derived autologous primary myeloma cells compared with mock T cells (Fig. 2f, g), underscoring the appreciable anti-MM efficacy of FcRH5 CAR-T cells in the autologous settings ex vivo. Moreover, almost all the patient-derived primary myeloma cells were killed by autologous FcRH5 CAR-T cells following overnight co-culture (Supplementary Fig. 5a), corroborating that the remaining myeloma cells following short-term co-culture were less likely to represent "resistant" myeloma cells. Furthermore, primary myeloma cells from a newly-diagnosed patient as well as a relapsed patient lacking BCMA

expression while retaining abundant FcRH5 expression, which were largely refractory to recognition and lysis by autologous BCMA CAR-T cells containing the same scFv in the CAR structure as that currently demonstrating potent clinical activity against MM[2–4], remained highly susceptible to recognition and elimination by autologous FcRH5 CAR-T cells (Supplementary Fig. 5b–d).

## FcRH5 CAR-T cells showed potent anti-MM efficiency against xenografts
We proceeded to evaluate the anti-MM potential of FcRH5 CAR-T cells in an NCI-H929 s.c. xenograft model established by

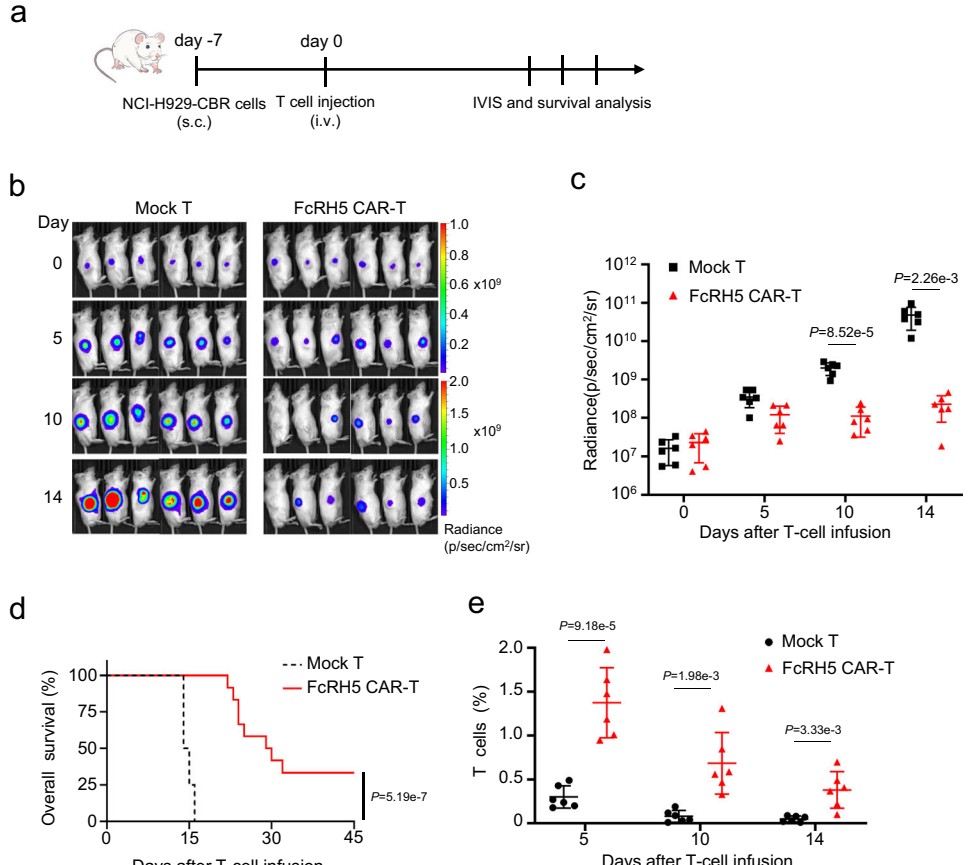

**Fig. 3 | FcRH5 CAR-T cells display tumoricidal activity in the subcutaneous NCI-H929-CBR xenograft model. a** Experimental schematic: Male NOG mice aged 6–8 weeks were subcutaneously inoculated with $5 \times 10^6$ NCI-H929-CBR cells expressing click beetle red (CBR) luciferase on day −7, and were then intravenously infused with $5 \times 10^6$ mock T or FcRH5 CAR-T cells from two different donors ($n = 6$ mice per group for each donor) when the tumors became palpable (day 0). **b** Representative tumor bioluminescence of NOG mice at different time points. **c** Graph showed the quantification of whole-body luminescence in NOG mice from each group at different time points with the lines connecting the mean values. Data shown are representative for results obtained in independent experiments with T-cell from 2 donors. Mean ± SD, $n = 6$ mice, unpaired two-tailed Student $t$ test, **d** Kaplan–Meier curve for the overall survival of mice from different treatment group ($n = 12$ mice per group). Log-rank (Mantel-Cox) test. **e** The percentage of T cells (gated on human CD45⁺CD3⁺) in peripheral blood of NOG mice at different time points was determined by flow cytometric analysis. Data are representative of two independent experiments with T-cell from 2 donors ($n = 12$ mice per group in total). Mean ± SD, unpaired two-tailed Student $t$ test,.

subcutaneously inoculating luciferase-expressing NCI-H929 cells into the flank of NOG mice (Fig. 3a). Infusion with FcRH5 CAR-T cells from two different donors could efficiently diminish the growth of luciferase-expressing NCI-H929 xenografts, as determined by IVIS imaging, and consequently prolonged the survival of the tumor-bearing mice. Importantly, we found that human CD3⁺ T cells could still be easily detected in the peripheral blood of the FcRH5 CAR-T group two to three weeks post infusion (Fig. 3b−e). In addition, infusion with two doses of FcRH5 CAR-T cells from a different donor suppressed subcutaneous tumor growth (Supplementary Fig. 6a−d). Moreover, infused FcRH5 CAR-T cells could apparently infiltrate into the subcutaneous tumor lesions, and retained their proliferative potential as manifested by positive staining of Ki67 Supplementary Fig. 6e, f).

Although it has been reported that in vitro-expanded NCI-H929 cells isolated from mice intravenous inoculation of NCI-H929 could be used to generate uniform disseminated tumors in a new batch of mice[22], we failed to generate such a disseminated MM xenograft model. As a result, we opted to evaluate the anti-tumor efficacy of FcRH5 CAR-T cells in a disseminated MM xenografts model established by intravenously inoculating luciferase-expressing MM.1s-FcRH5 cells into NOG mice (Fig. 4a). For side-by-side comparison of the activity of FcRH5 CAR with the BCMA CAR, a group of tumor-bearing mice were subject to therapy with BCMA CAR-T cells. We found that infusion with FcRH5 CAR-T or BCMA CAR-T cells from two different donors eliminated tumor lumps throughout the body, as shown by bioluminescence imaging (Fig. 4b, c), which effectively translated to prolonged mouse survival time (Fig. 4d). Moreover, human CD3⁺ T cells could be evidently discernible in the peripheral blood of FcRH5 CAR-T group or BCMA CAR-T group even three weeks post infusion (Fig. 4e). In an independent experiment, we confirmed that treatment with two doses of either CAR-T cells from a different donor evoke the remarkable anti-tumor efficacy (Supplementary Fig. 7a−d), and flow cytometric analysis of the mononuclear cells in the peripheral blood, spine as well as bone marrow of the mice showed that human CD138⁺ myeloma cells were almost eliminated while human CD3⁺ T cells were evidently present at these sites three weeks following treatment (Supplementary Fig. 7e, f). Furthermore, positive Ki67 staining of the residual CAR-T cells in spine and bone marrow of the mice suggested that infused CAR-T cells maintained the expansive capacity (Supplementary Fig. 7g, h).

Next, we examined whether FcRH5 CAR-T cells could exert therapeutic effect in a BCMA antigen deficiency murine xenograft model, in which NCI-H929^BCMA-KO cells (Supplementary Fig. 8a) were subcutaneously injected into the flanks of NOG mice. In vitro functional characterization showed that FcRH5 CAR-T cells were capable of

 5

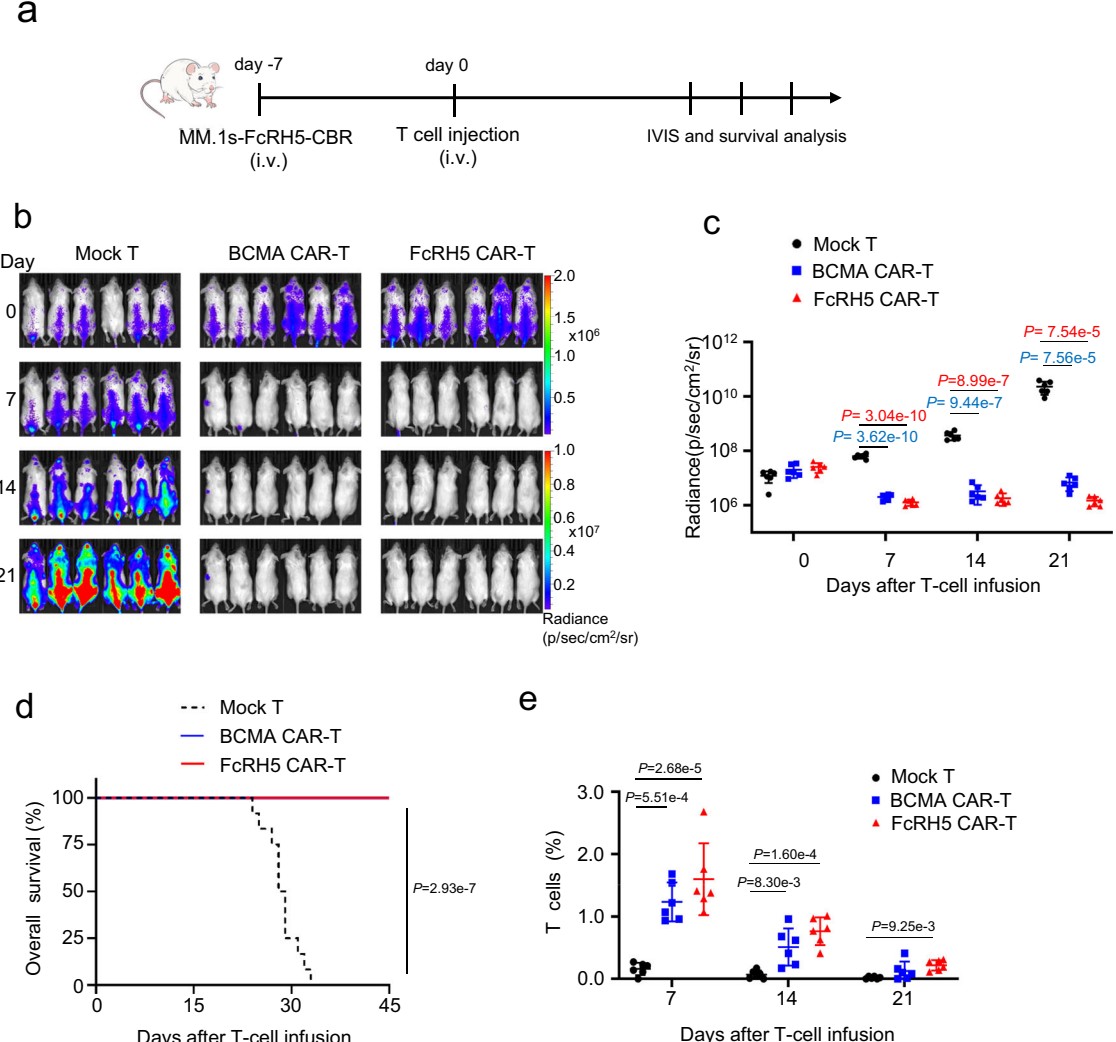

**Fig. 4 | FcRH5 CAR-T cells induce durable tumor regression in a disseminated MM xenograft model. a** Experimental schematic: Male NOG mice aged 6–8 weeks were intravenously inoculated with $5 \times 10^6$ MM.1s-FcRH5-CBR cells on day −7, and were then treated with $5 \times 10^6$ mock T, FcRH5 CAR-T cells or BCMA CAR-T cells from two different donors ($n = 6$ mice per group for each donor) on day 0. **b** Representative tumor bioluminescence of NOG mice at different time points. **c** Graph showed the quantification of whole-body luminescence in NOG mice from each group at different time points with the lines connecting means. Data shown are representative for results obtained in independent experiments with T-cell

from 2 donors. Mean ± SD, $n = 6$ mice, One-way ANOVA with Dunnett's correction for multiple comparison. **d** Kaplan–Meier curve for the overall survival of mice from different treatment group ($n = 12$ mice per group). Log-rank (Mantel–Cox) test. **e** The percentage of T cells (gated on human CD45[+]CD3[+]) in peripheral blood of NOG mice was determined by flow cytometric analysis. Data are representative of two independent experiments with T-cell from 2 donors ($n = 12$ mice per group in total). Mean ± SD, One-way ANOVA with Dunnett's correction for multiple comparison.

triggering robust cytotoxicity and releasing large amounts of IFN-γ and IL-2 upon stimulation with NCI-H929[BCMA-KO] cells, while BCMA CAR-T cells were more similar to mock T cells (Supplementary Fig. 8b–d). Interestingly, adoptive transfer of FcRH5 CAR-T cells from two different donors elicited remarkable tumoricidal effects in the mice bearing subcutaneous NCI-H929[BCMA-KO] xenografts, contrasting with uncontrolled tumor growth in the mice receiving BCMA CAR-T cells (Supplementary Fig. 8e–g).

**Influence of sFcRH5 on the functionality of FcRH5 CAR-T cells**
Malignant plasma cells from MM patients secrete large amounts of soluble FCRH5 (sFcRH5) protein into the circulation, which may mainly exist in two different forms[23,24] (Fig. 5a). The predominant form is secreted IRTA2a protein encoded by alternative splicing product of FcRH5, and the other is shed membrane IRTA2c (sIRTA2c) protein predicted to form by cleavage of membrane-tethered FcRH5 near the cell membrane by cellular proteinases[23]. To address

whether recognition of target cells by FcRH5 CAR-T cells could be abrogated by two forms of sFcRH5, cytolysis and cytokine production were quantified in response to escalating doses of exogenous recombinant sFcRH5 proteins. Recognition and lysis of NCI-H929 cells by FcRH5 CAR-T cells was not hampered by secreted IRTA2a protein in concentrations of over 10-fold higher than the median FcRH5 protein levels found in the serum of MM patients[24], as indicated by equivalent 6-hour cytotoxicity and IFN-γ secretion (Fig. 5b, c). In contrast, the 6-hour cytotoxicity of FcRH5 CAR-T cells was impaired in the presence of sIRTA2c protein at a concentration just above 500 ng/ml (Fig. 5d), which could be largely rescued by either increasing the E/T ratio or extending the co-culture time (Fig. 5e, f). In addition, sIRTA2c protein alone could evoke IFN-γ secretion by FcRH5 CAR-T cells, suggesting that it could be directly engaged by the latter (Fig. 5g). These findings suggested that distinct forms of sFcRH5 proteins variably affected the effector function of FcRH5 CAR-T cells.

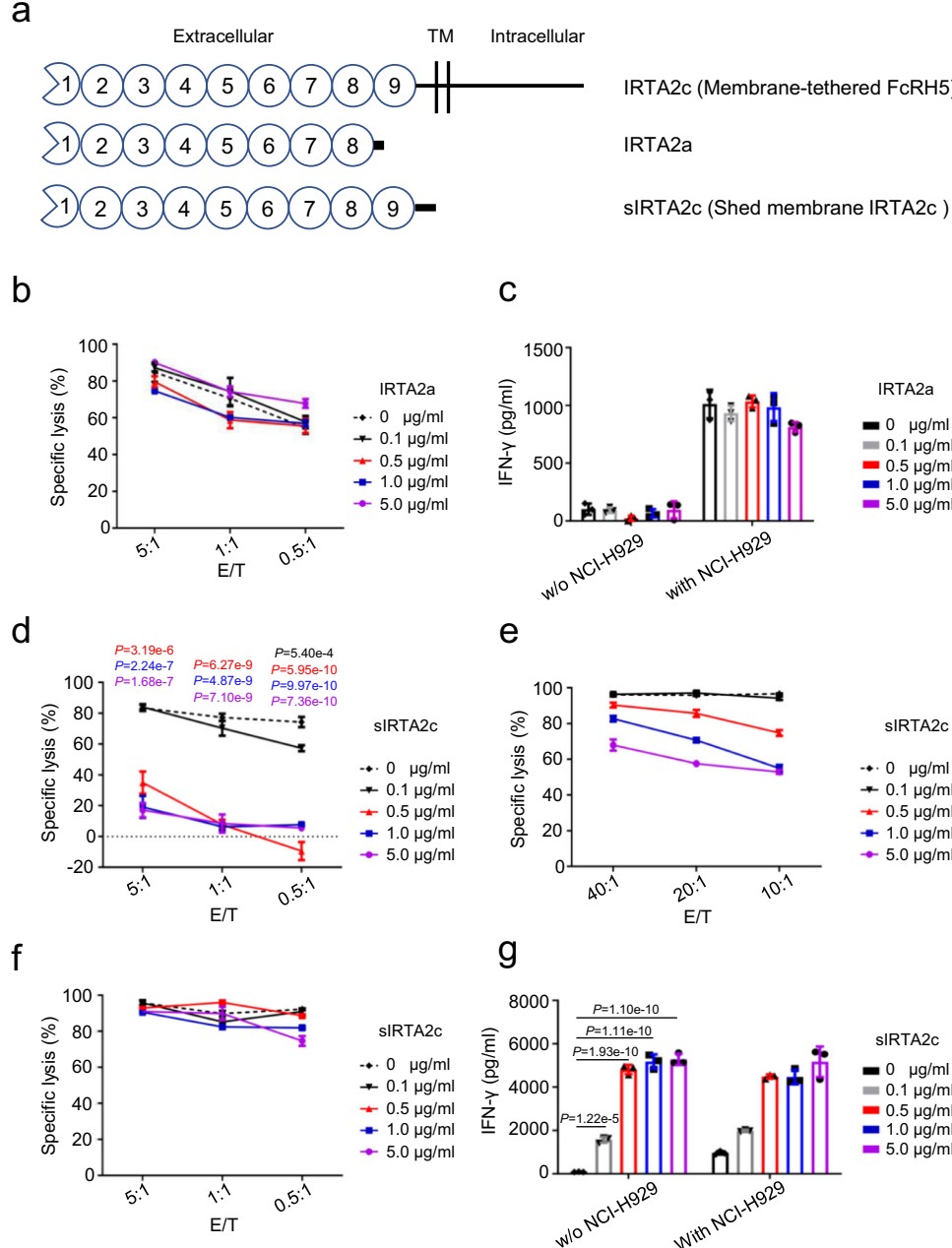

**Fig. 5 | Two distinct forms of soluble FcRH5 (sFcRH5) proteins disparately influence the effector function of FcRH5 CAR-T cells. a** Schematic representation of membrane-tethered FcRH5 (IRTA2c) protein and two distinct forms of sFcRH5 proteins including IRTA2a and sIRTA2c. **b** The cytolytic activity of FcRH5 CAR-T cells against NCI-H929 myeloma cells at various effector: target cell (E/T) ratios in the presence of grading concentrations of recombinant IRTA2a protein was determined by a 6-h luciferase-based cytolytic assay. Mean ± SD, $n = 3$ independent co-cultures. **c** FcRH5 CAR-T cells were stimulated with grading concentrations of recombinant IRTA2a protein in the presence or absence of NCI-H929 target cells for 24 h, and cell-free supernatants were harvested for evaluating IFN-γ secretion by ELISA. Mean ± SD, $n = 3$ independent co-cultures. **d** The cytolytic activity of FcRH5 CAR-T cells against NCI-H929 myeloma cells at the E/T ratios ranging from 5:1 to 0.5:1 in the presence of grading concentration of recombinant sIRTA2c protein was determined by the 6-h luciferase-based cytolytic assay. Mean ± SD. $n = 3$

independent co-cultures, one-way ANOVA with post hoc Tukey-Kramer test. **e** The cytolytic activity of FcRH5 CAR-T cells against NCI-H929 myeloma cells at the E/T ratios ranging from 40:1 to 10:1 in the presence of grading concentration of recombinant sIRTA2c protein were determined by a 6-h luciferase-based cytolytic assay. Mean ± SD, $n = 3$ independent co-cultures, one-way ANOVA with post hoc Tukey-Kramer test. **f** The cytolytic activity of FcRH5 CAR-T cells against NCI-H929 myeloma cells at low E/T ratios ranging from 5:1 to 0.5:1 in the presence of grading concentrations of recombinant sIRTA2c protein were determined by a 24-h luciferase-based cytolytic assay. Mean ± SD, $n = 3$ independent co-cultures. **g** FcRH5 CAR-T cells were stimulated with grading concentrations of recombinant sIRTA2c protein in the presence or absence of NCI-H929 target cells for 24 h, and cell-free supernatants were harvested for evaluating IFN-γ secretion by ELISA. Mean ± SD, $n = 3$ independent co-cultures, one-way ANOVA with post hoc Tukey-Kramer test. Experiments of (**b**–**g**) were repeated with two different T cell donors.

## Development and characterization of FcRH5-BCMA bispecific CAR -T cells

Manipulation of T cells to express bispecific CARs capable of dually targeting two different antigens have been recognized as a promising

strategy to reduce the risk of antigen escape. We constructed the bispecific FcRH5/BCMA CAR and BCMA/FcRH5 CAR by connecting anti-FcRH5 scFv and anti-BCMA scFv in tandem in different orders in the CD28 signaling moiety-containing second generation vector

(Supplementary Fig. 9a). Two different tandem CAR (TanCAR) constructs were transduced into bulk T cells, and the surface expression of TanCAR extracellular domain on the resultant bispecific CAR-T cells was detected using protein L staining. As illustrated in Supplementary Fig. 9b, FcRH5/BCMA CAR T cells were prominently positive for protein L staining indicative of surface localization of the TanCAR in its entirety, while BCMA/FcRH5 CAR-T cells were incapable to react with protein L, reflecting poor surface expression of TanCAR. In accordance with TanCAR expression pattern, FcRH5/BCMA CAR-T cells showed obvious cytotoxicity and IFN-γsecretion against target cells expressing FcRH5 and BCMA alone or in combination (Supplementary Fig. 9c–f), while BCMA/FcRH5 CAR-T cells failed to respond properly and were therefore excluded for further characterization. Compared with mono-specific CAR-T cells, FcRH5/BCMA CAR-T cells showed equivalent or even superior cytotoxicity and cytokine-secreting capacity (Fig. 6a, b). Of note, the CD4 v.s. CD8 composition did not differ significantly among mock T, mono-specific CAR-T cells and bispecific CAR-T cells across the donors (Supplementary Fig. 10a, b).

Lastly, we evaluated the efficacy of FcRH5/BCMA CAR-T cells in a subcutaneous NCI-H929 xenograft model (Fig. 6c). We observed that infusion with mono-specific or FcRH5/BCMA CAR-T cells from two different donors inhibited the growth of subcutaneous luciferase-labelled NCI-H929 xenografts and prolonged the survival of the mice (Fig. 6d–f); and more importantly, the FcRH5/BCMA CAR-T group displayed a longer median survival time and a higher percentage of human CD3$^+$ T cells in the circulation than either of the mono-specific CAR-T groups (Fig.6f, g). In addition, treatment with two doses of mono-specific CAR-T cells or FcRH5/BCMA CAR-T cells from a different donor led to tumor regression, as evidenced by reduced tumor weights at the end of the experiment endpoint (Supplementary Fig. 11a–c). In addition, a higher percentage of human CD3$^+$ T cells were noted in the tumors from FcRH5/BCMA CAR-T cell-treated group than those from either mono-specific CAR-T cell-treated group (Supplementary Fig. 11d, e), reflective of improved tumoral infiltration.

## Discussion

In this study, we sought to determine whether FcRH5, a surface antigen highly expressed in malignant plasma cells could be used as a potential target for CAR-T cells against MM maintaining or lacking BCMA antigen. We observed both FcRH5 and BCMA proteins were highly but variably expressed on the CD138$^+$ malignant plasma cells from 28 MM patients. In addition, for each patient, the expression levels of FcRH5 and BCMA appeared independent of one another. Particularly, we noted that some MM patients with abundant FCRH5 expression only have negligible levels of BCMA expression, implying that FcRH5 could be preferentially targeted in these patients. One limitation of this study lies in that a more precise quantitative analysis of antigen expression (e.g. number of molecules/cell) was not performed when characterizing the surface expression of BCMA and FcRH5 by flow cytometry analysis.

We then developed FcRH5 CAR-T cells and demonstrated that FcRH5 CAR-T cells mediated equivalent anti-tumor efficacy against MM cells or other target cells co-expressing FcRH5 and BCMA in comparison to BCMA CAR-T cells both in vitro and in two widely used MM xenograft murine models. Of note, both FcRH5 CAR-T and BCMA CAR-T cells could efficiently infiltrate into tumor sites and proliferate. These findings suggest that adoptive transfer of FcRH5 CAR-T cells may achieve potent and durable anti-tumor efficacy analogous to immunotherapy with BCMA CAR-T cells. Furthermore, we found NCI-H929 cells with BCMA knockout became unresponsive to BCMA CAR-T cells, yet remained highly susceptible to recognition and killing by FcRH5 CAR-T cells as demonstrated both in vitro and in the xenograft tumor model. Moreover, the anti-tumor capacity of FcRH5 CAR-T cells against patient-derived primary MM cells irrespective of BCMA expression was validated in the autologous setting, closely mimicking

a clinical scenario. These lines of evidence provide a strong rationale for evaluating FcRH5 CAR-T cells in treating MM with low or dim BCMA expression at presentation or alternatively relapsing with down-regulation or loss of BCMA following BCMA-targeted immunotherapies.

An important issue concerning the CAR-T therapy against FcRH5 is the potentially inhibitory impact of sFcRH5 antigens on the ability to target membrane-bound FcRH5, particularly in MM patients with high levels of sFcRH5 present in the circulation, as reported by others[24]. Due to the technical limitations, namely a lack of appropriate specific antibody, we could not set up an ELISA system for the detection of sFcRH5 in the circulation of MM patients. Nevertheless, we examined the potential impact of two types of sFcRH5 including IRTA2a and sIRTA2c on the effector function of FcRH5 CAR-T cells and found that IRTA2a, even at very high concentrations, did not impact on the effector function of FcRH5 CAR-T cells, whereas sIRTA2c at the concentration as low as 500 ng/ml could apparently impair the short-term cytotoxicity of FcRH5 CAR-T cells. Coupled with apparent IFN-γ release by FcRH5 CAR-T cells upon stimulation with sIRTA2c alone, we hypothesize that sIRTA2c protein directly bound FcRH5 CAR-T cells and consequently interfered with the interactions between FcRH5 CAR-T cells and surface FcRH5 on MM cells within a short period of time. Intriguingly, this inhibitory effect could be largely abolished when we prolonged the incubation time or increased the E/T ratio, hinting that the negative impact was transient and reversible, thereby allowing CAR-reengagement of membrane-tethered FcRH5 antigen on MM cells. On the other hand, it has been recognized that the predominant population of sFcRH5 is most likely to be IRTA2a rather than sIRTA2c, which was corroborated by the fact that 293T cells transfected with the IRTA2a, but not IRTA2c expression construct, produced sFcRH5 at very high levels in the supernatants[23]. Based on the aforementioned evidence, we believe the existence of two forms of sFcRH5 in the circulation of patients should not become a hurdle to the development of immunotherapy with FcRH5-specific CAR-T cells.

It is essential that target antigens selected for CAR-T therapy are not expressed on the surface of normal essential cells[25]. Early studies demonstrated that FcRH5 was preferentially expressed by B lineage cells in the spleen, tonsils, and lymph nodes by Nothernblot analysis[26]. It has been documented recently that FcRH5 mRNA was selectively expressed in B-lineage cells based on the analysis of GTEx sample set, consisting of 8,555 samples from 544 donors over 53 tissues[18]. Examination of the transcriptomics data of 54 human normal tissues from the Consensus datasets (the Human Protein Atlas) demonstrated that FcRH5 transcript was predominantly expressed in some lymphoid tissues including tonsil, lymph node, spleen and appendix, and was also detected at a much lower level in various gastrointestinal organs together with urinary bladder, salivary gland, and thymus, while its expression in the remaining tissues or organs was low or undetectable (Supplementary Fig. 12a). Moreover, inspection of single cell RNA sequencing (scRNAseq) data from 29 human normal tissue and peripheral blood mononuclear cell samples consisting of 79 cell types revealed that analogous to CD19 and BCMA, FcRH5 mRNA was highly and exclusively expressed in plasma cells and B cells, as well as in the early spermatids to a lesser extent (Supplementary Fig. 12b). Therefore, the mRNA signal detected in the above-mentioned tissues or organs was very likely derived from infiltrating B cells and plasma cells. However, due to lack of the anti-FcRH5 antibody suitable for immunohistochemical (IHC) staining, there is still no data available regarding the detailed expression profile of FcRH5 protein on human normal tissue, representing an important unresolved question. In spite of this, the data from the ongoing phase-I study (NCT03275103) of cevostamab, a FcRH5xCD3 bispecific antibody (BsAb) monotherapy in a large cohort of patients with heavily pre-treated RRMM, demonstrated promising activity with durable response and manageable safety without causing serious damage to any essential organ or tissue[27],

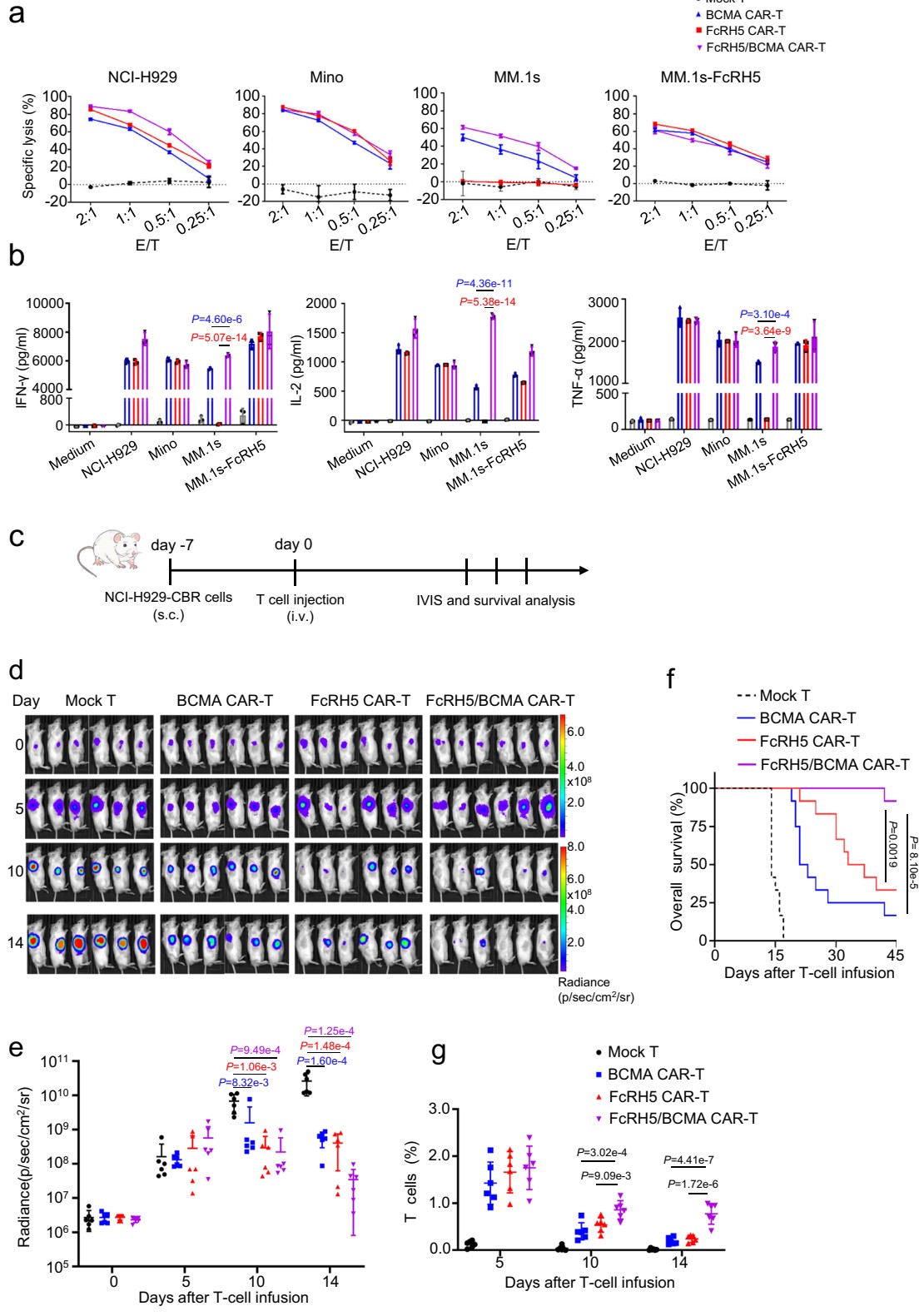

which may at least partly ease this safety concern. In addition, we verified that FcRH5 protein was almost absent on the surface of essential hematopoietic subsets including T cells, NK cells, monocytes and CD34+ HPSCs, but still expressed on B cells albeit at very low levels (Supplementary Fig. 13a). In agreement with this, FcRH5 CAR-T cells recognized B cells rather than the other four subsets as determined by an IFN-γ release assay (Supplementary Fig. 13b). Interestingly, only the minimal cytotoxicity against B cells was observed in FcRH5 CAR-T cells

even after the long-term co-culture (Supplementary Fig. 13c, d), demonstrating that FcRH5 CAR-T cells merely elicited limited damage to a minor portion of normal B cells. Similar to FcRH5 CAR-T cells, FcRH5/BCMA CAR-T cells could secrete abundant IFN-γ upon co-culture with B cells while merely displaying the minimal cytotoxicity (Supplementary Fig. 13e, f). In addition, infusion of supplemental immunoglobulins can be adopted in case of serious hypogammaglo-bulinemia due to the damage to normal plasma cells and B cells, which

**Fig. 6 | FcRH5/BCMA bispecific CAR-T cells demonstrate robust anti-MM capacity both in vitro and in vivo. a** Cytolytic activity of bispecific CAR-T cells and mono-specific CAR-T cells against indicated target cells was determined by the 6-h luciferase-based cytolytic assay. Mean ± SD, $n = 3$ independent co-cultures. **b** Bispecific CAR-T cells or mono-specific CAR-T cells were co-incubated with indicated target cells for 24 h, and cell-free supernatants were harvested for evaluating IFN-γ, IL-2 and TNF-α secretion. Mean ± SD, $n = 3$ independent co-cultures. Experiments of (**a**, **b**) were repeated with two different T cell donors. **c** Experimental schematic: Male NOG mice aged 6–8 weeks mice were subcutaneously inoculated with $5 \times 10^6$ NCI-H929-CBR cells expressing click beetle red (CBR) luciferase on day −7, and were then intravenously infused with $5 \times 10^6$ mock T, FcRH5 CAR-T, BCMA CAR-T or FcRH5/BCMA CAR-T cells from two different donors ($n = 6$ mice per group for each donor) on day 0 when the tumors became palpable. **d** Representative tumor bioluminescence of NOG mice at different time points. **e** Graph showed the quantification of whole-body luminescence in NOG mice from each group at different time points with the lines connecting means. Data shown are representative for results obtained in independent experiments with T-cell from 2 donors. Mean ± SD, $n = 6$ mice per group, One-way ANOVA with Dunnett's correction for multiple comparison. **f** Kaplan–Meier curve for the overall survival of mice from different treatment group ($n = 12$ mice per group). Log-rank (Mantel-Cox) test. **g** The percentage of T cells (gated on human CD45[+]CD3[+]) in peripheral blood of NOG mice at different time points was determined by flow cytometric analysis. Data are representative of two independent experiments with T-cell from 2 donors ($n = 12$ mice per group in total). Mean ± SD, One-way ANOVA with Dunnett's correction for multiple comparison.

has been demonstrated in the patients receiving BCMA CAR-T cells[28,29]. Therefore, FcRH5 expression on normal plasma cells as well as some normal B cells will not hamper therapeutically targeting FcRH5 with CAR-T cells. Moreover, to mitigate any potential or even unexpected toxicities of FcRH5 CAR-T cells, we demonstrated the feasibility of incorporating the clinically validated iC9 suicide gene system as a molecular switch (Supplementary Fig. 14a–d) as described previously[30].

Antigen escape has been observed in the relapsed patients following immunotherapy with CAR-T cells targeting different antigens such as CD19, CD22 and BCMA[31–35], hinting that it is not unique to any single antigen but reflects the evolution of resistance shared by various types of malignant cells under targeted therapy-imposed selective pressures[36]. Therefore, we hypothesize that downregulation or loss of FcRH5 is still likely to occur in MM patients following immunotherapy with FcRH5 CAR-T cells. One feasible strategy to overcome this obstacle is to endow T cells with the capacity to simultaneously engage different surface antigens, which can be implemented by expression of tandem CAR (dual-scFv single-stalk CAR) or bicistronic CAR (single input CARs encoded in bicistronic casettes)[37,38] on T cells. In this study, we designed TanCARs incorporating anti-FcRH5 scFv and anti-BCMA scFv connected in different orientation. Of the two configurations of TanCARs, only FcRH5/BCMA CAR appropriately expressed the Tan-CAR exodomain and therefore functioned properly when encountering target cells expressing FcRH5 and BCMA alone or in combination, underscoring the essential role of optimizing scFv orientation in development of fully functional TanCAR. Furthermore, we demonstrated that FcRH5/BCMA TanCAR-T cells displayed superior antitumor capacity in the subcutaneous NCI-H929 xenograft model than mono-specific CAR-T cells, which could be translated into the significantly improved survival of the mice, coinciding with improved persistence and intratumoral infiltration of T cells in this group. This finding was in agreement with a previous report regarding tandem CD19/CD20 CAR-T cells[39]. Currently, identifying the optimal approaches of simultaneously engaging multiple antigens has become a crucial goal of further CAR-T research[40–43]. It will be intriguing to identify the optimal ways of dually targeting FcRH5 and BCMA with CAR T cells in order to achieve the potent anti-MM efficacy while diminishing the likelihood of relapse from antigen loss.

Other than MM, multiple B cell malignancies such as hairy cell leukemia, chronic lymphocytic leukemia and mantle cell lymphoma frequently express surface FcRH5 on malignant cells[23,24,44], broadening the spectrum of hematological malignancies amendable to immunotherapy with FcRH5 CAR-T cells. Indeed, the capability of FcRH5 CAR-T cells to efficiently recognize and lyse FcRH5[+] mantle lymphoma cell line Mino has been illustrated in present study, and we plan to systemically evaluate the anti-tumor efficacy of FcRH5 CAR-T cells in numerous B-cell malignancies. Our clinical translation of immunotherapy targeting FcRH5, either alone or in combination with BCMA, in advanced MM naïve or resistant to BCMA-targeted therapy is currently underway.

## Methods

### Cell lines and culture condition
All human cell lines were obtained from the American Type Culture Collection (ATCC). Human multiple myeloma cell lines NCI-H929 and MM.1s as well as human mantle cell lymphoma cell line Mino were maintained in RPMI 1640 medium supplemented with 10% fetal bovine serum (FBS). Human embryonic kidney 293T cells were cultured in DMEM medium containing 10% FBS. Stable expression of FcRH5, firefly luciferase (FFL) or click beetle red (CBR) luciferase in various cell lines was performed by lentiviral transduction[30] and the cloning primers sequences were listed in Supplementary Table 1.

### Human participants
Human peripheral blood and bone marrow from healthy normal donors or MM patients were obtained in accordance with the Declaration of Helsinki after collecting written informed consent to participate in research protocols approved by the Faculty Hospital Ethics Committee at the First Affiliated Hospital of Soochow University (Suzhou, China). Human peripheral blood mononuclear cells (PBMCs) or bone marrow mononuclear cells (BMMCs) were isolated using Lymphoprep (STEMCELL Technologies) via density gradient centrifugation. Primary CD138[+] myeloma cells were positively selected from bone marrow aspirates of patients using human anti-CD138 MicroBeads and magnet-assisted cell sorting (MACS, Miltenyi Biotech), according to the manufacturer's instruction. All the samples were from Hematological Biobank, Jiangsu Biobank of Clinical Resources.

### Construction of lentiviral vectors
To generate anti-FcRH5 or anti-BCMA CAR expression constructs, the codon-optimized DNA sequence encoding either anti-FcRH5 scFv derived from 1G7 antibody[18] or anti-BCMA scFv from C11D5.3 antibody[1] with a $(G_4S)_3$ linker to separate VH and VL chains were synthesized and ligated into the lentiviral construct designated PCDH-SFFV-28z-IRES.EGFP containing a CD8 hinge, CD28 transmembrane and intracellular domain as well as CD3ξ intracellular domain as we previously described[30]. Bispecific FcRH5-BCMA CAR constructs were generated by replacing the anti-FcRH5 scFv portion in the anti-FcRH5 CAR construct with the DNA fragment encoding anti-FcRH5 scFv and anti-BCMA scFv connected in tandem in alternative orientation, linked by $(EAAAK)_3$, and the schematic CAR structure was illustrated in Fig. 6a. To generate a bicistronic expression construct containing the induced caspase 9 (iC9) suicide gene and anti-FcRH5 CAR, the iC9-encoding sequence was fused in frame with the sequence encoding T2A self-cleaved peptide and subsequently subcloned into the aforementioned anti-FcRH5 CAR expression vector upstream of anti-FcRH5 CAR cassette. The constructs encoding anti-FcRH5 specific CAR can be only available for the non-commercial use and for pre-clinical studies upon reasonable request from the corresponding authors, and a relevant material transfer agreement (MTA) may be needed.

For the generation of the lentiviral constructs expressing FcRH5, FFL and CBR, their corresponding coding domain sequences (CDS)

were amplified and ligated into PCDH-SFFV-IRES.EGFP or PLVX-EF1α-IRES.EGFP following enzymatic digestion. The sequences of cloning primers used in the study can be found in Supplementary Table 1.

## Lentivirus production

The lentiviral expression constructs together with the packaging plasmids psPax2 and PMD2.G (Addgene) were co-transfected into 293T cells[30]. The viral supernatants were harvested 48 h post-transfection, filtered through a PVDF filter and stored at −80 °C freezer.

## Transduction of human T cells

Peripheral blood mononuclear cells (PBMCs) from healthy donors or MM patients were stimulated with ultra-LEAFTM purified anti-Human CD3 antibody (Biolegend, catalog number 317326, clone OKT3, 1 μg/ml, dilution:1:2500) and ultra-LEAFTM purified anti-Human CD28 antibody (Biolegend, catalog number 302934, clone CD28.2, 1 μg/ml, dilution:1:2500) in RPMI-1640 medium containing 10% FBS and 250 IU/mL of interleukin-2 (IL-2) for 48 h[30]. The bulk T cells were resuspended in the virus solution containing 6 μg/ml polybrene and centrifuged at $600 \times g$ for 1 h, then transferred into a $CO_2$ incubator. 6 h later, virus solution was replaced with fresh medium containing 250 IU/mL IL-2. The second round of infection was performed on the next day. For the downstream experiments, we adjusted the number of T cells based on the transduction efficiency or according to the percentage of CAR-positive cells.

## Western blot analysis

Cell lysates were separated by SDS-PAGE gel and transferred to nitrocellulose membrane. The membrane was probed with mouse anti-human CD3ζ mAb (Santa Cruz, catalog number sc-166435, clone: E3, dilution:1:100) or GAPDH mAb (Biolegend, catalog number 649203, clone:FF26A/F9)) followed by incubation with a horseradish peroxidase–conjugated goat anti-mouse IgG antibody (Santa Cruz, catalog number sc-516102, dilution:1:1000)[30]. Antibody binding was detected using an enhanced chemiluminescence reagent (GE Healthcare Biosciences).

## Flow cytometry analysis

BMMCs were stained with anti-CD138 (BD Biosciences, catalog number 552026, clone:MI15, dilution:1:200) mAb to identify malignant plasma cells. To monitor expression of FcRH5 and BCMA on the surface of MM cells, the cells were stained with anti-FcRH5 mAb (Biolegend, catalog number 340306, clone:509F6, dilution:1:200) or anti-BCMA mAb (Biolegend, catalog number 357505, clone:19F2, dilution:1:200), PBMCs or mobilized PBMCs were stained with fluorochrome-conjugated antibodies specific for CD3 (Biolegend, catalog number 300412, clone:UCHT1, dilution:1:200), CD8 (Biolegend, catalog number 344718, colne:RPA-T8, dilution:1:200), CD14 (Biolegend, catalog number 301804, clone:M5E2, dilution:1:200), CD19 (Biolegend, catalog number 363006, clone:SJ25C1, dilution:1:200), CD56 (BD Bioscience, catalog number 557747, clone:B159, dilution:1:200) or CD34 (Miltenyi Biotec, catalog number 130-098-139, clone:AC136, dilution:1:200) to identify different subsets. The surface expression of anti-FcRH5 CAR on transduced T cells was determined by Protein L staining[45]. Briefly, T cells were incubated in biotinylated protein L (GenScript, catalog number M00097, 1 μg/ml) for 45 min, washed twice, and then detected with allophycocyanin (APC)-conjugated streptavidin (Jackson ImmunoResearch, catalog number 016-130-084, dilution:1:200). For detection of anti-BCMA CAR on the surface, T cells were washed with PBS containing 4% bovine serum albumin (BSA), incubated with biotin-labeled goat anti-mouse IgG F(ab')₂ polyclonal antibody (Jackson ImmunoResearch, catalog number 115-066-072, dilution:1:200) or normal polyclonal goat IgG antibody (Jackson ImmunoResearch, catalog number 005-060-003, dilution:1:200), then stained with APC-conjugated

streptavidin (Jackson ImmunoResearch, catalog number 016-130-084, dilution:1:200). For determination of the small molecule dimerizer drug AP1903-induced apoptosis, transduced T cells were treated with either DMSO or 10 nM AP1903 (MedChemExpress) for 24 h, followed by staining with Annexin V-APC (Biolegend, catalog number 640920, dilution:1:500) and 7-AAD (Biolegend, catalog number 559925, dilution:1:500) according to the manufacturer's protocol. Antibody staining was monitored with a Novocyte flow cytometer equipped with the NovoExpress v1.5.0 software. Data analysis was performed using FlowJo v10.6.2 software (Tree Star Inc.).

## Cytotoxicity assays

The luciferase-based cytotoxicity was used for evaluation of T cell-mediated lysis of various cell lines[30]. Target cells expressing FFL or CBR luciferase were co-cultured with T cells at various effector cells/target cells (E/T) ratios. Target cells alone were cultured as the control. When indicated, serial dilutions of soluble recombinant FcRH5 proteins were also added into the cocultures. At the indicated time points, the Bright-Glo luciferase reagent (Promega) was added into the wells, and the luminescence value was recorded using a luminescence microplate reader. The percentage of specific lysis was calculated as: (the luminescence values of target cells alone−luminescence values of target cell co-cultured with T cells) / luminescence values of target cells alone ×100%.

For assessment of T cell-mediated lysis of patient-derived MM cells, CD138+ primary myeloma cells were isolated from BMMC with CD138 microbeads (Miltenyi Biotech) and labeled with cell proliferation dye eFluor-670 (Thermo Fisher Scientific), and then the labeled cell were cultured alone or co-cultured with T cells for the indicated time period. Afterwards, 123count eBeads counting beads (Thermo Fisher Scientific) were added into the cultures and the absolute number of eFluor670-labeled target cells was quantified by flow cytometry analysis according to the manufacture's protocol. The percentage of specific lysis was calculated as: (the absolute number of target cells when cultured alone−the absolute number of target cells when co-cultured with T cells)/ the absolute number of target cells when cultured alone × 100%.

## Cytokine release assay

Target cells were co-cultured with an equal number of effector cells in triplicate in the presence of graded concentrations of soluble recombinant FcRH5 proteins in 96-well U-bottom plates in complete culture medium without cytokines at 37 °C for 24 h. Cell-free supernatants were harvested following centrifugation and assessed for secretion of IFN-γ, IL-2 and TNF-α using corresponding ELISA kits (Biolegend) according to the manufacturer's protocol.

## eFlour-670 cell proliferation assay

T cells were washed with PBS and labeled with eFluor-670 (Biolegend). Labeled T cells were then co-cultured with an equal number of irradiated target cells for 3 or 5 days in RP1640 medium supplemented with 10% FBS and 10IU/mL IL-2[30], and eFluor-670 dilution was monitored by flow cytometry.

## CD107a degranulation assay

The degranulation assay was performed as previously described[30] with minor modifications. Target cells were co-cultured with an equal number of T cells for 1 h. APC-conjugated anti-CD107a (BD Bioscience, catalog number 560664, clone:H4A3, dilution:1:200) or IgG1 isotype antibody (BD Biosciences, catalog number 554681, clone: MOPC-21, dilution:1:200) in addition to monensin (BD Biosciences) were then added for 4 h at 37 °C. The cells were subsequently washed with PBS and stained with CD69-BV421 (BD Bioscience, catalog number 562884, clone:FN50, dilution:1:200) and analyzed on a Novocyte flow cytometer.

## Intracellular staining of granzyme B

T cells were cocultured with an equal number of target cells for 6 h, and then labeled with BV421 conjugated anti-CD3 antibody (BD Biosciences, catalog number 563798, clone:UCHT1, dilution:1:200), followed by permeabilization and intracellular staining with Alexa Fluor 647-conjugated anti-Granzyme-B antibody (BD Biosciences, catalog number 560212, clone:GB11, dilution:1:200) using the Cytofix/Cytoperm kit (BD Biosciences)[30].

## CRISPR/Cas9-mediated BCMA gene knockout

For BCMA knockout, the single guide RNA (sgRNA) targeting BCMA (GAAGAACATCGAAGTTGACA) was cloned into the modified lentiCRISPR v2 vector (Addgene plasmid 52961) with Puro$^R$ cassette replaced with EGFP according to the standard protocol provided on Addgene website, and the sgRNA primer sequences used were listed in Supplementary Table 1. Then the resultant plasmid was cotransfected with the packaging plasmids psPax2 and pMD2.G into 293T cells to make lentiviruses used to transduce NCI-H929 cells. Following transduction, GFP-positive cells were sorted by flow sorter and expanded as single clones. Individual colonies were then expanded and stained with anti-BCMA antibody (Biolegend, catalog number 357505, clone:19F2, dilution:1:200), and only the clones with deficient surface BCMA expression were used for further study.

## Preparation of two forms of soluble FcRH5 proteins

The extracellular portion (Gln16-Arg844) of membrane-bound FcRH5 protein with a C-terminal 6 His-tag representative of shed membrane IRTA2c protein was purchased directly from R&D Company (Catalogue number: 2078-FC), and the secreted IRTA2a protein with a C-terminal 6 His-tag was produced by 293T cells transfected with the plasmid encoding IRTA2a in the supernatants using Ni-NTA affinity purification system. The concentration of recombinant IRTA2a protein was determined using a Bradford assay.

## In vivo xenograft mouse models

All animal experiments were approved by the Institutional Laboratory Animal Care and Use Committee of Soochow University (Suzhou, China). NOD.Cg-*Prkdc*$^{scid}$*Il2rg*$^{tm1Sug}$/JicCrl (NOG) mice (6–8 weeks of age) were purchased from Vital River Laboratories (Beijing, China) and maintained at the Animal Facility of Soochow University with a 12 h light/dark cycle and a temperature range of 20–22 °C and humidity range 40–70%. For the subcutaneous xenograft model, male NOG mice were intradermally inoculated with $5 \times 10^6$ NCI-H929 cells or NCI-H929$^{BCMA-KO}$ cells[46]. The longest dimension (L) of tumors and width perpendicular to the length (W) were measured with caliper every 2 days, and the tumor volumes were calculated per the formula $1/2 \times L \times W^2$. When the tumors became palpable with the volume of approximately 100 mm$^3$, the mice were randomly allocated into four groups and intravenously infused with mock T, FcRH5 CAR-T, BCMA CAR-T or FcRH5/BCMA bispecific CAR-T cells normalized for transduction efficiency. For the indicated experiments, the mice received a second dose of T cells two days later. For the survival analysis, the mice were euthanized when the tumor length reached 15 mm. The maximal tumor size permitted by the Institutional Laboratory Animal Care and Use Committee was 18 mm in diameter, and this size was not exceeded. Tumors were harvested and ground for flow cytometric analysis as indicated. Immunostaining with anti-human Ki67 antibody (Biolegend, catalog number 350514, clone:Ki-67, dilution:1:200) was performed by fixing and permeabilizing the cells with the Foxp3/transcription Factor Staining Kit (eBioscience).

A disseminated MM model was established by intravenous injection of MM cells into male NOG mice as we previously described[13]. NOG mice were intravenously inoculated with $5 \times 10^6$ MM.1s-FcRH5-FFL cells, and then randomly allocated to treatment groups administered with mock T, FcRH5 CAR-T or BCMA CAR-T cells normalized for transduction efficiency via tail vein injection as indicated. For monitoring tumor burden, the mice were intraperitoneally infused with D-luciferin (150 mg/kg body weight; Gold Biotechnology), anesthetized with isoflurane, and imaged on the In Vivo Imaging System (IVIS) with Living Image software (PerkinElmer). Mice were euthanized when moribund or determined by a veterinarian. For some experiments, mice were euthanized at the indicated time points, and mononuclear cells isolated from peripheral blood, spine or bone marrow were used for flow cytometry.

## Statistical analysis

Statistical significance was evaluated using unpaired two-tailed Student's *t*-test, One-way ANOVA or two-way ANOVA as described in the figure legends. ANOVA followed by Tukey's, Šídák's, Dunnett's post hoc tests or Bonferroni correction were used for multi-group comparisons. Linear regression analysis was performed to assess the correlation between FcRH5 expression and BCMA expression in MM patients. Survival curves were constructed using the Kaplan-Meier method and statistical analyses of survival was performed using a log-rank (Mantel-Cox) test. Results were considered statistically significant when P or adjusted $P < 0.05$. Graphpad Prism software v8.0 was used for statistical analyses and graph generation.

## Reporting summary

Further information on research design is available in the Nature Portfolio Reporting Summary linked to this article.

## Data availability

Analyses of FcRH5, CD19 or BCMA transcript expression in human normal tissue or cells were based on publicly available data from The Human Protein Atlas, which can be accessed using the following links: https://www.proteinatlas.org/ENSG00000143297-FCRL5/tissue; https://www.proteinatlas.org/ENSG00000143297-FCRL5/single+cell+type; https://www.proteinatlas.org/ENSG00000177455-CD19/single+cell+type; https://www.proteinatlas.org/ENSG00000048462-TNFRSF17/single+cell+type. All other data supporting the findings of this study are available in the article and its Supplementary Information file. Source data are provided with this paper.

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

## Acknowledgements

This work was supported by the grants from the National Key R&D Program of China (2022YFC2502700 (Y.X.), 2019YFC0840604 (D.W.), 2017YFA0104502 (D.W.), National Natural Science Foundation of China (81770216 (J.C.), 82020108003 (D.W.), 81730003 (D.W.), 82200204 (D.J.)), Translational Research Grant of NCRCH (2020WSB04 (J.C.)), Priority Academic Program Development of Jiangsu Higher Education Institutions (PAPD), Key R&D Program of Jiangsu Province (BE2019798

(D.W.)), Natural Science Foundation of Jiangsu Province (BK20220248 (D.W.)), Jiangsu Medical Outstanding Talents Project (JCRCA2016002 (D.W.)), Jiangsu Provincial Key Medical Center (YXZXA2016002 (D.W.)) and Suzhou Science and Technology Program Project (SLT201911 (D.W.)). The Hematologic Biobank, Jiangsu Biobank of Clinical Resources was acknowledged for providing the clinical samples.

## Author contributions

D.J., H.H. and H.Q. designed and performed experiments and analyzed data. Y.G., X.S. and K.T. designed, performed experiments and analyzed data. T.Z., Y.Z., X.T., J.F., W.Q. and W.C. contributed to the data analysis and interpretation of the results. J.C., D.W. and Y.X. designed experiments, analyzed data, wrote the manuscript and supervised the project.

## Competing interests

J.C., D.W., Y.X., T.Z. and D.J. hold patents and/or patent applications in the fields of cancer immunotherapy. The remaining authors declare no competing interests.
