## [Peer Review File · Nature Communications]

REVIEWER COMMENTS

Reviewer #1 (Remarks to the Author):

The authors in this paper show that targeting

FcRH5 with CAR-T cells represents a promising immunotherapy in multiple myeloma. The paper is innovative and well done

-The noteworthy results are the demonstration that FcRH5/BCMA bispecific CAR-T cells efficiently recognize MM cells expressing FcRH5 and/or BCMA, and induce substantial tumor regression in vivo

-The work is significant to the field and related fields. The work is original.

-The work supports the conclusions and claims, any additional evidence needed.

-The methodology meets the expected standards in this field and enough detail is provided in the methods for the work to be reproduced.

I request additional experiments to check whether FcRH5/BCMA bispecific CAR-T cells may have an effect on normal cells including B cells and stem cells. I suggest to evaluate the expression profile of FcRH5 in hematopoietic stem cells and normal B cells.

Reviewer #2 (Remarks to the Author):

The majority of CARs developed for multiple myeloma (MM) has been focused on targeting B-cell maturation antigen (BCMA). Recently, one product targeting BCMA was approved by the FDA for r/r MM. Antigen loss or downregulation after BCMA CAR T cell therapy has been observed in several MM patients – therefore development of CARs targeting other or additional antigens on MM cells are warranted. The manuscript submitted by Dongpeng et al., addresses this important drawback by designing a novel CAR against FcRH5 which is expressed on MM cells.

Figure 1: The study is initiated by investigation of BCMA and FcRH5 expression on CD138+ MM cells from 28 patients and MM cell lines. Authors find % wise higher expression of FcRH5 compared to BCMA on patient cells. Correlation analysis between BCMA and FcRH5 expression on patient cells are non-significant but there seems to be a clear tendency. Authors also provide evidence of some patients retaining FcRH5 expression while BCMA expression is lost.

Comments: There seems to be a tendency that patients with low BCMA expression also have low FcRH5 expression although not significant.

Figure 2: Authors develop a second generation CAR composed of an anti FcRH5 scFv, linked to a CD8 hinge, and a CD28 transmembrane domain, CD28 co-stim domain and CD3 signaling domain. Using lentiviral transduction human primary T cells are modified to express the aFcRH5 CAR. Cells induce significantly higher tumor lysis compared to mock transduced T cells in a dose-dependent manner. Likewise, CAR T cells produce relevant cytokines (IL2, TNF α and IFN γ) stimulation with irradiated tumor cells. Neither killing nor cytokine production is observed in co-cultures with FcRH5-negative cell lines. Proliferation of CAR T cells is observed in co-cultures with FcRH5+ cell lines. Authors claim this effect is due to synapse formation (Fig 2e) because they have an increase in double+ CD138/CD3 cells – increase from 2.6 to 10%. Authors perform experiment with autologous CAR T cells against tumor and show significant IFN γ release and killing compared to mock transduced cells.

Comments: Experiments are done in biological triplicates and two donors have been used. The “synapse” formation experiment is weak and unnecessary and just confuses the reader. I suggest to take this out. In vitro evaluation is sufficient in my opinion. Using autologous CAR T cells against patient derived tumor is good and more representative than initial killing assays.

Figure 3: Authors perform an in vivo mouse model using 5e6 NCI-H929 tumor cells. They provide 5e6 CAR T cells or mock T cells 2 and 5 days after tumor injection (6 mice/condition). Mice go until day 14 before being sacrificed. They observe a significant reduction in tumor burden (capillary measurements) and tumor volume

Comments: Highly artificial setting when such high dose of CAR is provided 2 days after “engraftment”? Tumor burden close to 0 seems from figure. It seems in vivo experiments are based on a CAR and mock T cells from a single donor. I don't agree this is sufficiently solid to argue an effect.

Figure 4: Authors perform an additional in vivo model using 5e6 MM.1s (BCMA+ naturally) transduced with FcRH5. Mice are injected with 5e6 tumor cells 7 days prior to CAR administration

(dose 5e6 CARs or Mock T cells). Mice are dosed twice day 0 and 2. Both CAR treatment groups are cured with prolonged survival until around day 5. There is a significant difference between overall survival between BCMA and FcRH5 CAR treated mice (however very subtle). In a separate experiment, authors sacrifice mice 3 weeks post CAR injection and find significant reduction of CD138+ myeloma cells in BCMA or FcRH5 CAR treated mice compared to mock. At this time significant more T cells are also found in BCMA or FcRH5 CAR treated mice compared to mock indicating CAR T cells have undergone proliferation as response to meeting tumor cells. Ki67 staining shows that CAR T cells maintained the expansive capacity.

Comments: Highly artificial setting – with high dose of CAR T cells (overall 10e6) are given over 2 days. Only 6 mice pr. Group reflects the use of one donor. In my opinion this is not sufficient for a mouse study which should be based on at least 10 donors. They claim a better survival of FcRH5 CAR treated mice compared to BCMA CAR T cells. Why is this? Because the tumor grows out or just a coincidence after such long time (GVHD) – this should be commented on. It would be preferable to quantify the flux combined with images. (Of note – we have had multiple problems with MM1s cells – they have generated a very high response from many donors and we have seen that mock T cells cure the tumor within 3 weeks).

Authors provide evidence that FcRH5 CAR T cells can control tumor growth in a “BCMA-antigen escape model” where BCMA have been knocked out using CRISPR/Cas9. This data is presented in supplementary.

Figure 5: It is known that some tumor antigens can be cleaved of the tumor cells (e.g. BCMA) which can hinder the efficacy of CAR T cell therapy both in terms of dim tumors that are not recognized or shedded sAg binding to circulation CARs so they are unable to bind tumor cells. Like BCMA, FcRH5 can also be shed from tumors and may cause problems for FcRH5 CAR T cells. FcRH5 can be shed from MM cells in two variants IRTA2a and sIRTA2c. Authors perform a series of killing assays where CAR-T cells efficacy against NCI-H929 myeloma in co-culture systems with 0 – 5 ug/ml IRTA2a and sIRTA2c supplemented. They find that FcRH5 CAR T cells retain their killing capacity in assays with IRTA2a but there is a dose dependent decline in tumor killing capacity of FcRH5 CAR T cells when sIRTA2c is supplemented (at second dose FcRH5 CAR T cell efficacy is lost. This effect is abolished when the effector:target ratio is increased or co-culture is prolonged from 6 to 24h.

Comments: CAR efficacy abolished by soluble antigens have been observed for other antigens (e.g. BCMA, CD19). sFcRH5 have been found circulating in high levels in MM patients (stated in ref 24). Authors state it is now recognized that IRTA2a is the predominant form of shed FcRH5. This version did not seem to affect the FcRH5 CAR T cells and therefor authors hypothesize that sFcRH5 would be a hurdle for FcRH5 CAR T cell development.

Figure 6: As antigen-escape is a major obstacle for the curative potential of CAR T cell therapy with ag- negative relapse observed for e.g. BCMA, CD19, CD22 CARs. One reason for this might be selective pressure exerted by a single targeting CAR. Therefor authors developed a tandem CAR, where the CAR is designed as a single molecule with the BCMA and FcRH5 scfv's connected in tandem by a flexible linker fused to a Cd8 hinge domain, Cd28 co-stim and CD3z domain. Two constructs were made with a difference in the scfv orientation (FcRH5 – linker - BCMA scfv or BCMA – linker – FcRH5). The FcRH5 – linker - BCMA scfv configuration was found to be superior and able to respond to both BCMA and FcRH5 single positive cell lines and was further evaluated. The FcRH5/BCMA tandem CAR showed equal tumor killing capacity and cytokine production to single targeting BCMA or FcRH5 CAR T cells. Tandem CARs were tested in vivo against 5e6 NCI-H929 (BCMA+ and FcRH5+). Single targeting BCMA or FcRH5 CAR T cells or mock T cells were used as control. Authors observed a significant reduction in tumor volume for all CAR T cells compared to mock. Higher CD3+ T cells were found in the tumors of tandem CAR treated mice compared to single targeting CARs indicative of improved tumoral infiltration.

Comments: Mice experiments seems to be conducted with 6 mice receiving cells for the same donor. Experiments with multiple donors are preferred. Too high a dose of CAR T cells to evaluate any real difference between groups.

Overall: Overall the manuscript submitted by Dongpeng et al, addresses an important drawback of current CAR T cell therapy for MM by designing a novel CAR against FcRH5 as alternative to currently approved BCMA CAR T cells. Overall the manuscript is well written with experiments well executed and repetitions. The in vivo experiments are performed with a significantly high amount of CAR T cells injected not representing actual circumstances for human use. With such high doses it is very difficult to access the actual functionality of the FcRH5 CAR alone. In addition, potential

differences between mono-specific and tandem CAR T cells are lost in such system. Likewise, it seems all in vivo experiments are conducted with T cells from a single donor – at least two donors should be included for each in vivo model to better recapitulate natural donor-to-donor variants. The manuscript would benefit from repeating in vivo experiments with more donors and including lower doses of CAR T cells injected to fully validate the in vivo efficacy of these novel CARs.

Reviewer #3 (Remarks to the Author):

This is an interesting manuscript with pre-clinical data on FcRH5 as a target for CAR T cells in multiple myeloma. The data set and scope of the manuscript is broad, but requires further refinement and elaboration in order to provide a balanced and informative perspective for the reader.

Major comments:

1. Expression of FcRH5 on primary myeloma cells and cell lines: The data shown in Fig.1 imply the FcRH5 is present non a higher percentage of myeloma cells compared to BCMA. However, BCMA seems to be lower than reported in previous studies. Histogram overlay are not an appropriate way for determining the percentage of myeloma cells that express a target antigen and authors ought to revisit their work flow, and add quantitative data (i.e. molecules/cell) for both BCMA and FcRH5. Further, authors ought to add data on FcRH5 expression on myeloma cells obtained from bone marrow vs. extramedullar lesions; and different genetic aberrations.
2. In vitro analyses with FcRH5 CAR T cells: The authors ought to clarify the proportions of CD8 vs. CD4 CAR-expressing T cells, and whether these was equivalent in FcRH5 CAR and BCMA CAR in their experiments. The conclusion that FcRH5 CAR T cells form immune synapses with MM cells is not supported by data (a cluster assay by flow cytometry is not adequate). In the lysis assay against primary in vitro, a substantial proportion of targets cells is left at the end of the assay. Does killing reach 100% with further follow up, or are there 'resistant' myeloma cells?
3. In vivo analyses with FcRH5 CAR T cells: Reporting on the NCI-H929 s.c. myeloma models is quite limited and should include longer follow-up, BLI analyses, survival curve, CAR T pharmacokinetic at multiple points, etc. The same myeloma cell line should be used for the disseminated myeloma model. The use of MM1.s cells that overexpress FcRH5 is not adequate, in particular for a comparison to BCMA CAR T cells (BCMA is expressed 'only' at physiologic levels in this cell line and not overexpressed). Also, only 1 dose of CAR T cells should be administered in the disseminated model.
4. Soluble FcRH5: This is an interesting aspect and data that show the concentrations of both sFcRH5 isoforms in MM patients vs. healthy donors should be included. Also, data on the concentrations of both sFcRH5 isoforms in the s.c. and i.v. MM mouse models. It would be interesting to know if there is increased cytokine production by FcRH5 CAR T cells in vivo due to binding of sFcRH5 to the CAR; and whether any of the bispecific CARs is less prone to binding either of the two sFcRH5 isoforms.
5. Safety profile and requirement for a suicide gene: Expression of FcRH5 on normal B cells is an acceptable off-tumor reactivity and may not require including a suicide gene into T cells. Can the authors provide addtl. data on FcRH5 expression in other normal tissues, and ideally cross-reactivity studies with their binding domain to demonstrate absence of recognition of normal tissues?
6. Language and style: it is advisable to go through a round of language editing to enhance clarity for reader.

RESPONSE TO REVIEWERS' COMMENTS

Reviewer #1 (Remarks to the Author):

The authors in this paper show that targeting FcRH5 with CAR-T cells represents a promising immunotherapy in multiple myeloma. The paper is innovative and well done -The noteworthy results are the demonstration that FcRH5/BCMA bispecific CAR-T cells efficiently recognize MM cells expressing FcRH5 and/or BCMA, and induce substantial tumor regression in vivo

-The work is significance to the field and related fields. The work is original.

- the work support the conclusions and claims, any additional evidence needed.

-The methodology meet the expected standards in this field and enough detail provided in the methods for the work to be reproduced.

I request additional experiments to check whether FcRH5/BCMA bispecific CAR-T cells may have an effect of normal cells including B cells and stem cells. I suggest to evaluate the expression profile of FcRH5 in hemaopoietic stem cells and normal B cells.

Response: Thanks a lot for the reviewer's positive comments on our work. Per the reviewer's suggestion, we have checked the response of FCRH5/BCMA CAR-T cells to normal B cells and hematopoietic stem cells, and found that, similar to FCRH5 CAR-T cells, FCRH5/BCMA CAR-T cells could secrete abundant IFN- γ but display minimal cytolytic activity in response to normal B cells; and meanwhile, FCRH5/BCMA CAR-T cells had no obvious impact on the hematopoietic stem cells based on IFN- γ secretion assay and cytolytic assay (data shown in Supp. Figure 13e,f). In addition, the expression profile of FcRH5 on hematopoietic stem cells and normal B cells was evaluated by flow cytometry analysis, and the representative results were illustrated in Supp. Figure 13a, showing that FcRH5 was almost absent on hematopoietic stem cells but still dimly expressed on normal B cells.

Reviewer #2 (Remarks to the Author):

The majority of CARs developed for multiple myeloma (MM) has been focused on targeting B-cell maturation antigen (BCMA). Recently, one product targeting BCMA was approved by the FDA for r/r MM. Antigen loss or downregulation after BCMA CAR T cell therapy has been observed in several MM patients – therefor development of CARs targeting other or additional antigens on MM cells are warranted. The manuscript submitted by Dongpeng et al., addresses this important drawback by designing a novel CAR against FcRH5 which is expressed on MM cells.

Figure 1: The study is initiated by investigation BCMA and FcRH5 expression on CD138+ MM cells from 28 patients and MM cell lines. Authors find % wise higher expression of FcRH5 compared to BCMA on patient cells. Correlation analysis between BCMA and FcRG5 expression on patient cells are non-significant but there seems to be a clear tendency. Authors also provides evidence of some patients retaining FcRH5 expression while BCMA expression is lost.

Comments: There seems to be a tendency that patients with low BCMA expression also have low

FcRH5 expression although not significant.

Response: We would like to agree with the reviewer's comments on this point.

Figure 2: Authors develop a second generation CAR composed on an anti FcRH5 scfv, linked to a CD8 hinge, and a CD28 transmembrane domain, CD28 co-stim domain and CD3 signaling domain. Using lentiviral transduction human primary T cells are modified to express the aFcRH5 CAR. Cells induce significantly higher tumor lysis compared to mock transduced T cells in a dose-dependent manner. Likewise, CAR T cells produce relevant cytokines (IL2, TNF α and IFN γ) stimulation with irradiated tumor cells. Neither killing nor cytokine production is observed in co-cultures with FcRH5-negative cell lines. Proliferation of CAR T cells is observed in co-cultures with FcRH5+ cell lines. Authors claim this effect is due to synapse formation (Fig 2e) because they have an increase in double+ CD138/CD3 cells – increase from 2,6 to 10%. Authors perform experiment with autologous CAR T cells against tumor and show significant IFN γ release and killing compared to mock transduced cells.

Comments: Experiments is done in biological triplicates and two donors have been used. The “synapse” formation experiment is weak and unnecessary and just confuse the reader. I suggest to take this out. In vitro evaluation is sufficient in my opinion. Using autologous CAR T cells against patient derived tumor is good and more representative that initial killing assays.

Response: Thanks a lot for the reviewer's valuable comments and advice. Per your suggestion, we have deleted all the portion relevant to the “synapse” formation experiment.

Figure 3: Authors perform an in vivo mouse model using 5e6 NCI-H929 tumor cells. They provide 5e6 CAR T cells or mock T cells 2 and 5 days after tumor injection (6 mice/condition). Mice go until day 14 before being sacrificed. They observe a significant reduction in tumor burden (caplilar measurements) and tumor volume

Comments: Highly artificial setting when such high dose of CAR is provided 2 days after “engraftment”? Tumor burden close to 0 seems from figure. It seems in vivo experiments are based on a CAR and mock T cells from a single donor. I don't agree this is sufficiently solid to argue an effect.

Response: Thanks a lot for the reviewer's valuable comments and suggestion. In this experiment, NOG mice were subcutaneously inoculated with 5×10^6 NCI-H929 cells on day -7, and were then intravenously infused with 5×10^6 mock T or FcRH5 CAR-T cells on day 0 and day 2, meaning that CAR-T cells were administered 7 and 9 days after tumor cell engraftment (instead of 2 and 5 days after engraftment), respectively. The first dose of CAR-T cells were given when the tumors became palpable with the volume of approximately 100 mm³ in this experiment. By the way, we have not found any previous report about utilization of CAR-T cells to treat the NCI-H929 subcutaneous xenograft, while noticed one article (Nature Communications, (2020)11:283.) reporting treatment of RPMI8226 (a different myeloma cell line) subcutaneous xenograft with BCMA CAR-T cells when the tumor volume reached 25 or 50 mm³ according to the two graphs presented. To address the reviewer's concern about the dosage of CAR-T cells, we have set up two new mouse experiments with the luciferase-labelled NCI-H929 tumor-bearing mice treated with

only a single dose (5e6/mouse) of CAR-T cells (transduction efficiency is around 50%) from two different donors, with the data added in Figure 3a-h. By the way, treatment with a lower dose (2.5e6/mouse) of FcRH5 CAR-T cells could not effectively diminish the growth of subcutaneous NCI-H929 xenografts based on the result of our preliminary experiment as below.

Note: 5e6 luciferase-expressing NCI-H929 cells were subcutaneously inoculated into NOG mice, 7 days later (day 0), the mice were treated with 2.5e6 Mock T or FcRH5 CAR-T cells, and then tumor volumes were measured at the indicated time points (a) and the images of tumor nodules (b) were acquired on day 14.

Figure 4: Authors perform an additional in vivo model using 5e6 MM.1s (BCMA+ naturally) transduced with FcRH5. Mice are injected with 5e6 tumor cells 7 days prior to CAR administration (dose 5e6 CARs or Mock T cells). Mice are dosed twice day 0 and 2. Both CAR treatment groups are cured with prolonged survival until around day 5. There is a significant difference between overall survival between BCMA and FcRH5 CAR treated mice (however very subtle). In a separate experiment, authors sacrifice mice 3 weeks post CAR injection and find significant reduction of CD138+ myeloma cells in BCMA or FcRH5 CAR treated mice compared to mock. At this time significant more T cells are also found in BCMA or FcRH5 CAR treated mice compared to mock indicating CAR T cells have undergone proliferation as response to meeting tumor cells. Ki67 staining shows that CAR T cells maintained the expansive capacity.

Comments: Highly artificial setting – with high dose of CAR T cells (overall 10e6) are given over 2 days. Only 6 mice pr. Group reflects the use of one donor. In my opinion this is not sufficient for a mouse study which should be based on at least two donors. They claim a better survival of FcRH5 CAR treated mice compared to BCMA CAR T cells. Why is this? Because the tumor grows out or just a coincidence after such long time (GVHD) – this should be commented on. It would be preferable to quantify the flux combined with images. (Of note – we have had multiple problems with MM1s cells – they have generated a very high response from many donors and we have seen that mock T cells cure the tumor within 3 weeks).

Authors provide evidence that FcRH5 CAR T cells can control tumor growth in a “BCMA-antigen escape model” where BCMA have been knocked out using CRISPR/Cas9. This data is presented in supplementary.

Response: Thanks a lot for the reviewer's valuable comments and suggestion. Regarding the survival status of the mice treated with different CAR-T cells, although at the late stage one mice from BCMA CAR-T group died, there was no statistically significant difference ($P > 0.05$) in the overall survival time between two groups, and therefore we still can not claim a better survival of

FcRH5 CAR-T group compared to BCMA CAR-T group. During the period of experiment, we did not observe any obvious symptom of GVHD such as body weight loss, fur loss and hunched posture in the mice treated with either CAR-T cells, and therefore speculated that the one mouse from BCMA CAR-T group may die of tumor outgrowth. Moreover, to address the concern about the dosage of CAR-T cells as well as the number of donors used from this reviewer together with the suggestion from the reviewer 3 who claimed that only 1 dose of CAR T cells should be administered in the disseminated model, we have performed two new mouse experiments with the MM.1s-FcRH5 tumor-bearing mice treated with only a single dose (5×10^6 /mouse) of CAR-T cells (transduction efficiency is around 50%) from two different donors with the data added in Figure 4a-h, and also the graph showing quantitative analysis of tumor burden in terms of BLI have been included. In addition, we would like to thank the reviewer for sharing their experience on MM.1s mouse model with us, while based on our experience, we have not noticed the similar phenomenon for unknown reasons. Meanwhile, based on the result of our preliminary experiment, it seemed that treatment of MM.1s-FcRH5 tumor-bearing mice with BCMA CAR-T or FcRH5 CAR-T cells at the dosage of 2.5×10^6 /mouse could not efficiently control the growth of disseminated tumors.

Note: NOG mice were intravenously inoculated with 5×10^6 MM.1s-FcRH5-CBR cells on day -7, and were then treated with 2.5×10^6 mock T, FcRH5 CAR-T cells or BCMA CAR-T cells on day 0. BLI was performed to evaluate the tumor burden of NOG mice from each group at different time points.

Figure 5: It is known that some tumor antigens can be cleaved of the tumor cells (e.g. BCMA) which can hinder the efficacy of CAR T cell therapy both in terms of dim tumors that are not recognized or shedded sAg binding to circulation CARs so they are unable to bind tumor cells. Like BCMA, FcRH5 can also be shed from tumors and may cause problems for FcRH5 CAR T cells. FcRH5 can be shed from MM cells in two variants IRTA2a and sIRTA2c. Authors perform a series of killing assays where CAR-T cells efficacy against NCI-H929 myeloma in co-culture systems with 0 – 5 $\mu\text{g/ml}$ IRTA2a and sIRTA2c supplemented. They find that FcRH5 CAR T cells

retain their killing capacity in assays with IRTA2a but there is a dose dependent decline in tumor killing capacity of FcRH5 CAR T cells when sIRTA2c is supplemented (at second dose FcRH5 CAR T cell efficacy is lost. This effect is abolished when the effector : target ratio is increased or co-culture is prolonged from 6 to 24h.

Comments: CAR efficacy abolished by soluble antigens have been observed for other antigens (e.g. BCMA, CD19). sFcRH5 have been found circulating in high levels in MM patients (stated in ref 24). Authors state it is now recognized that IRTA2a is the predominant form of shed FcRH5. This version did not seem to affect the FcRH5 CAR T cells and therefore authors hypothesize that sFcRH5 would be a hurdle for FcRH5 CAR T cell development.

Response: Thanks a lot for the reviewer's valuable comments.

Figure 6: As antigen-escape is a major obstacle for the curative potential of CAR T cell therapy with an- negative relapse observed for e.g. BCMA, CD19, CD22 CARs. One reason for this might be selective pressure exerted by a single targeting CAR. Therefore authors developed a tandem CAR, where the CAR is designed as a single molecule with the BCMA and FcRH5 scfv's connected in tandem by a flexible linker fused to a Cd8 hinge domain, Cd28 co-stim and CD3z domain. Two constructs were made with a difference in the scfv orientation (FcRH5 – linker - BCMA scfv or BCMA – linker – FcRH5). The FcRH5 – linker - BCMA scfv configuration was found to be superior and able to respond to both BCMA and FcRH5 single positive cell lines and was further evaluated. The FcRH5/BCMA tandem CAR showed equal tumor killing capacity and cytokine production to single targeting BCMA or FcRH5 CAR T cells. Tandem CARs were tested in vivo against 5e6 NCI-H929 (BCMA+ and FcRH5+). Single targeting BCMA or FcRH5 CAR T cells or mock T cells were used as control. Authors observed a significant reduction in tumor volume for all CAR T cells compared to mock. Higher CD3+ T cells were found in the tumors of tandem CAR treated mice compared to single targeting CARs indicative of improved tumoral infiltration.

Comments: Mice experiments seems to be conducted with 6 mice receiving cells for the same donor. Experiments with multiple donors are preferred. Too high a dose of CAR T cells to evaluate any real difference between groups.

Response: Thanks a lot for the reviewer's valuable comments and suggestion. Per the reviewer's suggestion, we have set up two new mouse experiment using luciferase-labelled NCI-H929 cells in which the tumor-bearing mice were treated with a low dose (5e6/mouse) of mono-specific CAR-T cells or FcRH5/BCMA TanCAR-T cells from two different donors, and the data was added in Figure 6d-g and Supp. Figure 11b-e, indicating that FcRH5/BCMA TanCAR-T cells displayed a superior in vivo anti-MM efficacy than either of the mono-specific CAR-T cells.

Overall: Overall the manuscript submitted by Dongpeng et al, addresses an important drawback of current CAR T cell therapy for MM by designing a novel CAR against FcRH5 as alternative to currently approved BCMA CAR T cells. Overall the manuscript is well written with experiments well executed and repetitions. The in vivo experiments are performed with a significantly high amount of CAR T cells injected not representing actual circumstances for human use. With such high doses it is very difficult to access the actual functionality of the FcRH5 CAR alone. In

addition, potential differences between mono-specific and tandem CAR T cells are lost in such system. Likewise, it seems all in vivo experiments are conducted with T cells from a single donor – at least two donors should be included for each in vivo model to better recapitulate natural donor-to-donor variants. The manuscript would benefit from repeating in vivo experiments with more donors and including lower doses of CAR T cells injected to fully validate the in vivo efficacy of these novel CARs.

Response: Thanks a lot for the reviewer's valuable comments and suggestion. We have followed the suggestion from this reviewer and also the reviewer 3 to repeat all the in vivo experiments with only a single dose of CAR-T cells (5×10^6 /mouse) from two different donors with the data shown in Figures 3a-h, 4a-h, 6d-g and Supp. Figures 9a-d, 11b-e.

Reviewer #3 (Remarks to the Author):

This is an interesting manuscript with pre-clinical data on FcRH5 as a target for CAR T cells in multiple myeloma. The data set and scope of the manuscript is broad, but requires further refinement and elaboration in order to provide a balanced and informative perspective for the reader.

Major comments:

1. Expression of FcRH5 on primary myeloma cells and cell lines: The data shown in Fig.1 imply the FcRH5 is present non a higher percentage of myeloma cells compared to BCMA. However, BCMA seems to be lower than reported in previous studies. Histogram overlay are not an appropriate way for determining the percentage of myeloma cells that express a target antigen and authors ought to revisit their work flow, and add quantitative data (i.e. molecules/cell) for both BCMA and FcRH5. Further, authors ought to add data on FcRH5 expression on myeloma cells obtained from bone marrow vs. extramedullar lesions; and different genetic aberrations.

Response: Thanks a lot for the reviewer's valuable comments and suggestion. Expression of BCMA and FCRH5 in myeloma patients have been extensive characterized and reported in some studies, supporting that expression of both BCMA and FcRH5 appears to be quite variable across the myeloma patients. Please understand that numerous factors such as sample type, sample size, timing of sample collection, and laboratory technique as well as the specific antibody used may have affected the reporting of BCMA or FcRH5 levels.

We have quantified the expression levels of BCMA and FcRH5 in the form of so-called MFIR (the ratio of mean fluorescence intensity with specific antibody compared with isotype control), which has previously been adopted by others to quantify and compare expression of BCMA and TACI expression on the patient-derived myeloma cells [Blood (2011) 117 (3): 890–901.]. We have added the MFIR quantitative data shown in Fig. 1c in the updated manuscript for your reference. On the other hand, we know that Quantum™ Simply Cellular microspheres from Bangs laboratory are often used in the precise quantitative analysis of cellular antigen expression in the form of molecule/cell. Unfortunately, we failed to obtain the kit via the vendor and thus can not present the molecule/cell quantitative data as suggested.

We also agree that examining FcRH5 expression on myeloma cells obtained from bone marrow vs. extramedullar lesions will be of importance, which has not been reported previously. However,

freshly dissected extramedullary samples from myeloma patients are not available for us for some reasons including ethical consideration, and thus we do not have the chance to check FcRH5 expression on the fresh extramedullary samples. Meanwhile, since no well-recognized immunohistochemical (IHC) grade FcRH5-specific antibody is commercially available, we also can not evaluate FcRH5 expression on frozen or paraffin sections of human normal tissues. We feel very sorry for not being able to address this question at this stage.

Since FcRH5 gene is located in the chromosomal breakpoint in 1q21, 1q21 gain is the most relevant genetic aberration. It has been reported that there is a significant positive correlation between FcRH5 mRNA expression in myeloma cells and 1q21 gain in the patients (Cancer Cell. 2017 Mar 13;31(3):383-395.), yet whether FcRH5 protein level is elevated in the patients with 1q21 gain has not been addressed. We analyzed the FcRH5 protein expression in the myeloma patients with v.s. without 1q21 gain, and found that FcRH5 expression levels in terms of percentage of antigen positive cells or MFI were significantly elevated in the patients with 1q21 gain compared to those without 1q21 gain, and the data was added in Supp. Figure 1a,b.

2. In vitro analyses with FcRH5 CAR T cells: The authors ought to clarify the proportions of CD8 vs. CD4 CAR-expressing T cells, and whether these were equivalent in FcRH5 CAR and BCMA CAR in their experiments. The conclusion that FcRH5 CAR T cells form immune synapses with MM cells is not supported by data (a cluster assay by flow cytometry is not adequate). In the lysis assay against primary in vitro, a substantial proportion of target cells is left at the end of the assay. Does killing reach 100% with further follow up, or are there 'resistant' myeloma cells?

Response: Thanks a lot for the reviewer's valuable comments and suggestion. We have added the representative result shown in Supp. Figure 11a to support that the proportions of CD8⁺ v.s. CD4⁺ cells were equivalent in FcRH5 CAR-T and BCMA CAR-T cells. Considering that both reviewer 2 and 3 have questioned the value of the "synapse" formation experiment, and especially the reviewer 2 thought it was actually unnecessary and therefore suggested removing it, we followed the suggestion to delete all the relevant portion. Regarding the observation that still a substantial proportion of primary myeloma cells is left at the end of the short-term (6-h) cytolytic assay as shown in Fig. 2g, we know that CAR-T cells exert the cytolytic effect in a time-dependent manner, and thus extending the co-culture time may be expected to further increase the cytolytic effect. Actually, we have previously tested the long-term cytolytic assay and found that almost all the primary myeloma cells from the patient A, B and C were killed by autologous FcRH5 CAR-T cells following overnight (24-h) co-culture as shown below, although we did not include the data in the manuscript, corroborating that the remaining target cells following short-term co-culture were less likely to represent "resistant" myeloma cells.

Note: The patient-derived primary myeloma cells were labeled with eFlour-670 and co-cultured with autologous mock T or FcRH5 CAR-T cells at the E/T ratio of 5:1 for 24 hours, and then cytolytic effect of T cells was determined by a flow cytometry-based cytotoxicity assay.

3. In vivo analyses with FcRH5 CAR T cells: Reporting on the NCI-H929 s.c. myeloma models is quite limited and should include longer follow-up, BLI analyses, survival curve, CAR T pharmacokinetic at multiple points, etc. The same myeloma cell line should be used for the disseminated myeloma model. The use of MM1.s cells that overexpress FcRH5 is not adequate, in particular for a comparison to BCMA CAR T cells (BCMA is expressed 'only' at physiologic levels in this cell line and not overexpressed). Also, only 1 dose of CAR T cells should be administered in the disseminated model.

Response: Thanks a lot for the reviewer's comments and suggestion. We set up two new experiments in which luciferase-expressing NCI-H929 cells were inoculated into NOG mice prior to treatment with a single dose of CAR-T cells from two different donors so that we can monitor tumor burden by BLI analysis as suggested. Also in this report we have included longer follow-up, survival curve (note: the mice were sacrificed when the longest length reached 15 mm) and CAR-T pharmacokinetic at multiple time points, with the main data shown in Figure 3.

Based on our previous experience, NCI-H929 cells are generally not suitable for establishing a reliable disseminated myeloma model through intravenous injection possibly due to poor tumorigenicity. Our pilot study also showed that intravenous inoculation of up to 10e6 NCI-H929 cells failed to yield apparent disseminated tumor lesions in the majority of immunodeficient mice (n=6 mice) as determined by BLI after a couple of weeks as below. Therefore, we did not perform the test as suggested.

Note: Tumor lesions were monitored by BLI at the indicated time points after 10e6 luciferase-expressing NCI-H929 cells were intravenously inoculated into NOG mice.

As we mentioned in the manuscript, since MM1.s cells do not express detectable FcRH5 protein on the surface, we engineered MM.1s cells to express FcRH5 to a "physiological" level similar to that in primary myeloma cells from many patients on the surface. Actually, we have optimized the condition of viral infection and cell sorting to avoid overexpressing FcRH5 to a very high level. Given that MM1.s cells express BCMA protein at a physiological level per se, it appears not necessary to further manipulate its expression level. Moreover, the expression level of BCMA seems to be comparable to that of FcRH5 in the MM.1s-FcRH5 cells by flow cytometry analysis. Lastly, we have set up two new experiments with only a single dose of CAR-T cells from two different donors administered in the MM1.s-FcRH5 disseminated myeloma model, with the main data illustrated in Figure 4.

4. Soluble FcRH5: This is an interesting aspect and data that show the concentrations of both sFcRH5 isoforms in MM patients vs. healthy donors should be included. Also, data on the concentrations of both sFcRH5 isoforms in the s.c. and i.v. MM mouse models. It would be interesting to know if there is increased cytokine production by FcRH5 CAR T cells in vivo due to binding of sFcRH5 to the CAR; and whether any of the bispecific CARs is less prone to binding either of the two sFcRH5 isoforms.

Response: Thanks a lot for the reviewer's comments and suggestion. We noted that others have previously measured the levels of soluble FcRH5 in the blood of MM patients vs. healthy donors. Ise T et al. designed the sandwich ELISA which can detect both recombinant IRTA2a protein and recombinant shed IRTA2c protein (yet can not differentiate between them), and then used it to detect both sFcRH5 isoforms in the sera or plasma of MM patients vs. healthy donors, and reported that the soluble FcRH5 levels were <30-600 ng/ml with the median value of 188 ng/ml in the 193 healthy donors, vs. <30-11000 ng/ml with the median value of 481 ng/ml in 43 MM patients (*Leukemia, 2007;21:169-174*). We have cited the literature accordingly in the manuscript. Although we would like to apply this well-established ELISA system to measure soluble FcRH5 protein in MM patients, the FcRH5-specific antibodies used in the sandwich ELISA system are

not available for us. In addition, we have previously purchased two commercially available sFcRH5 ELISA kits but just found that both of them can only recognize the recombinant shed IRTA2c (sIRTA2c) protein rather than the recombinant IRTA2a protein. Due to lack of the appropriate ELISA system for detection of both sFcRH5 isoforms, we can not utilize them to measure the concentrations of both sFcRH5 isoforms in MM patients vs. healthy donors as suggested. For the same reason, we can not follow the reviewer's suggestion to evaluate the concentrations of both sFcRH5 isoforms in the s.c. and i.v. MM mouse models. By the way, we have evaluated the levels of sFcRH5 (sIRTA2c) in the supernatants of NCI-H929 cells and MM.1s-FcRH5 cells with this ELISA kit, and still no obviously detectable levels of sFcRH5 (sIRTA2c) was noticed, indicating that sIRTA2c may be present at very low levels or even be absent in the supernatants of both cell lines harboring a high amount of surface FcRH5 protein. Since both NCI-H929 and MM.1s-FcRH5 cells can not produce detectable levels of sFcRH5 (sIRTA2c), and also only sIRTA2c (rather than IRTA2a) can bind FcRH5 CAR-T cells and trigger cytokine secretion by FcRH5 CAR-T cells *in vitro*, we can not address the question regarding whether there is increased cytokine production by FcRH5 CAR-T cells in our animal models due to binding of sFcRH5 (presumably sIRTA2c) to the CAR, although it seems interesting. Once again, we would like to emphasize that, although theoretically there may exist two sFcRH5 isoforms in the circulation, the current experimental evidence strongly supports that IRTA2a is very likely to be the predominant form of sFcRH5, while there is still no convincing evidence to support the existence of sIRTA2c so far.

To address whether any of the bispecific CARs is less prone to binding either of the two sFcRH5 isoforms, we performed the IFN- γ ELISA assay and observed that BCMA/FcRH5 CAR-T cells did not produce detectable IFN- γ upon exposure to either of the sFcRH5 isoforms, and in contrast, FcRH5/BCMA CAR-T cells can produce abundant IFN- γ only upon stimulation with sIRTA2c, behaving like FcRH5 CAR-T cells. This indicated that FcRH5/BCMA CAR can bind sIRTA2c rather than IRTA2a, while BCMA/FcRH5 CAR, which actually did not display any effector function, can not bind either sFcRH5 isoform. The data was added in Supp. Figure 13g.

5. Safety profile and requirement for a suicide gene: Expression of FcRH5 on normal B cells is an acceptable off-tumor reactivity and may not require including a suicide gene into T cells. Can the authors provide addtl. data on FcRH5 expression in other normal tissues, and ideally cross-reactivity studies with their binding domain to demonstrate absence of recognition of normal tissues?

Response: Thanks a lot for the reviewer's valuable comments and suggestion. We believe that incorporation of a suicide gene may help minimize any risk of any unpredictable toxicity especially when we initiate the first clinical trial on FcRH5 CAR-T cells; and would like to consider omitting the suicide gene in the CAR construct in case that immunotherapy with FcRH5 CAR-T cells is proven to be safe enough. Regarding the question about expression pattern of FcRH5 in normal tissues other than B cells, we have analyzed FcRH5 protein expression on some key immune subsets including T cells (both resting and pre-activated), NK cells and monocytes together with hematopoietic stem/progenitor cells (HPSCs) by flow cytometry analysis, and found that FcRH5 protein was almost absent on all these subsets examined as shown in Supp. Figure 13a. In addition, it has been reported that FcRH5 mRNA was selectively expressed in B lineage cells

and tissues based on the analysis of GTEx sample set, consisting of 8555 samples from 544 donors over 53 tissues (Cancer Cell. 2017 Mar 13;31(3):383-395.). Unfortunately, there is still no putative IHC-grade anti-FcRH5 antibody available for detecting the expression of FcRH5 protein on the paraffin section or frozen section of human normal tissues. Moreover, since an FcRH5 ortholog does not exist in the mouse, we can not perform the cross-reactivity assay to investigate the possibility of recognition of any normal mouse tissue by FcRH5 CAR-T cells in our murine xenograft models.

6. Language and style: it is advisable to go through a round of language editing to enhance clarity for reader.

Response: Thanks a lot for the reviewer's suggestion. We have asked a native English speaker to help improve it. We would be very grateful if the reviewer would like to offer some detailed advice on the language editing for this updated manuscript.

REVIEWER COMMENTS

Reviewer #1 (expert in multiple myeloma, bone disease, CD38, cell survival, angiogenesis):

Authors responded to the issues of the reviewer

Reviewer #2 (expert in CAR T cells, bone marrow transplantation, haematological malignancies):

Overall, the manuscript reads well and is interesting and provides proof of concept for targeting FcRH5 on myeloma cells with CAR T cells.

The authors have addressed my previous comments.

Minor suggestions:

For the in vivo experiments, it would be helpful to include a schematic overview of the time line. I also suggest condensing figures 3 (and 4) and show representative images for a single donor and compile the numbers from both donors in single graph.

Reviewer #3 (expert in CAR T cells, bone marrow transplantation, haematological malignancies):

Overall, this is an interesting manuscript on FcRH5 as a possible target for CAR-T cell therapy in multiple myeloma. However, although the authors have revised their manuscript taking also the reviewers' comments into account, which is much appreciated, there are still further refinements that should be considered (see comments below) to provide a balanced perspective for the reader.

Re: 1. Expression of FcRH5 on primary myeloma cells and cell lines:

Thank you for the clarification. Regarding the BCMA and FcRH5 levels in MM patient samples: if data sets from MM patients are already available by others, this should be cited accordingly to put the data sets displayed in Fig. 1 of the revised manuscript into perspective. A quantitative data analysis of the expression (i.e. molecules/cell) is still highly recommended and should not be left out because a kit is not available. There are kits available also by other vendors (e.g. BD Biosciences™ Quantibrite™ Phycoerythrin (PE)-Beads) that could be employed for this purpose.

If there is no appropriate IHC FcRH5-specific antibody available to evaluate FcRH5 expression in human tissues: Could this data be extracted from the cancer genome atlas and the human protein atlas?

Re: Data displayed for the FcRH5 expression in patient samples with and without 1q21 mutations (Supp. Fig. 1 a,b) : Here, as for the flow cytometric data in Fig. 1, a quantitative analysis of the expression levels (i.e. molecules/cells) is recommended.

Re: 2. In vitro analyses with FcRH5 CAR T cells:

Thanks for including the data for the CD4 vs. CD8 composition of the CAR-T cells and the clarification on the cytotoxicity assay. However, this information should be clearly stated in the main text and/or appropriate figure legends, especially as most researchers in the field would raise the same question regarding the cytolytic activity of these CAR-T cell products (could be also added in the supplement). Furthermore, a summary of the CD4 vs. CD8 composition from all donors used in this study should be included so that it is clear that CAR T cells used for all experiments had a similar CD4:CD8 ratio.

Re: 3. In vivo analyses with FcRH5 CAR T cells:

Thanks for including follow-up, BLI analyses etc. for the NCI-H929 s.c. myeloma in the revised

manuscript. It is still highly recommended to use the same myeloma cell line for the disseminated myeloma model. It should be possible to establish such a model also with the NCI-H929 cell line as exemplified e.g. here: Mirandola et al. BMC Cancer 2011, 11:394 and here: Proc Natl Acad Sci USA. 2019 Mar 5;116(10):4592-4598. doi: 10.1073/pnas.1821733116.

Re: 4. Soluble FcRH5:

Thanks for the details, also for the bispecific CAR constructs. However, as others already established a sandwich ELISA to detect both isoforms, it would be highly recommended to inquire with the authors of this study if they could possibly share the antibodies and protocols to address this issue, especially for the detection of the isoforms in healthy vs. diseased states. This could also be interesting when using primary tumor material as targets instead of cell lines as this might also reveal which of the CAR constructs might be superior in a clinical setting.

Re: 5. Safety profile and requirement for a suicide gene

Thanks for the clarification why the suicide gene was added to the CAR construct. Re: expression data shown in Suppl. Fig. 13a: As already mentioned above: flow cytometric analysis without quantification is not sufficient to exclude possible low level expression of the target antigen on normal tissue. If mRNA data on expression of FcRH5 is already available, that should be referenced accordingly. Furthermore, as outlined above: if no suitable FcRH5 IHC antibody is available so far, please consider to check the cancer genome atlas and the human protein atlas if the data is available there. Re: cross-reactivity studies: Such a study should be performed on human tissue to demonstrate that the CAR construct does not recognize normal tissues.

RESPONSE TO REVIEWERS' COMMENTS

Reviewer #1 (Remarks to the Author):

Authors responded to the issues of the reviewer.

Response: We sincerely appreciate the reviewer's time and effort in improving our manuscript.

Reviewer #2 (Remarks to the Author):

Overall, the manuscript reads well and is interesting and provides proof of concept for targeting FcRH5 on myeloma cells with CAR T cells.

The authors have addressed my previous comments.

Minor suggestions:

For the in vivo experiments, it would be helpful to include a schematic overview of the time line. I also suggest condensing figures 3 (and 4) and show representative images for a single donor and compile the numbers from both donors in single graph.

Response: We are happy to know that the previous comments have been addressed and also are very thankful to the reviewer's valuable comments and advice. Per the reviewer's request, we have included a schematic overview of the time line for each in vivo experiment, condensed Figures 3, 4 as well as the original supplementary Figure 9 (the similar situation) with representative images for a single donor shown and also compiled the numbers from both donors in single graph. In addition, we noticed that the Figure 6 and the original supplementary Figure 11 displayed the parallel results from two different donors, and therefore combined both of them and showed the representative data similarly to ensure the consistence in the style for data presentation. We hope the reviewer would be satisfied with all those changes.

Reviewer #3 (Remarks to the Author):

Overall, this is an interesting manuscript on FcRH5 as a possible target for CAR-T cell therapy in multiple myeloma. However, although the authors have revised their manuscript taking also the reviewers' comments into account, which is much appreciated, there are still further refinements that should be considered (see comments below) to provide a balanced perspective for the reader.

Re: 1. Expression of FcRH5 on primary myeloma cells and cell lines:

Thank you for the clarification. Regarding the BCMA and FcRH5 levels in MM patient samples: if data sets from MM patients are already available by others, this should be cited accordingly to put the data sets displayed in Fig. 1 of the revised manuscript into perspective. A quantitative data analysis of the expression (i.e. molecules/cell) is still highly recommended and should not be left out because a kit is not available. There are kits available also by other vendors (e.g. BD Biosciences™ Quantibrite™ Phycoerythrin (PE)-Beads) that could be employed for this purpose.

Response: Thanks a lot for the comments and suggestion. We mentioned that others have previously characterized the surface expression of BCMA and FcRH5 proteins in MM patients and also cited the main relevant articles accordingly (Line 139) in the manuscript. Regarding the suggestion about quantification of FcRH5 in MM patient samples in the form of molecule/cell, even with the BD Biosciences™ Quantibrite™ Phycoerythrin (PE)-Beads, we still can not address this point given that the original patient samples actually ran out and also the fluorescent antibodies previously used for staining of FcRH5 and BCMA were of APC channel rather than PE channel. Please understand that the MM patient sample are really precious and actually difficult for us to obtain, and meanwhile, the sample volume is generally limited and also for a majority of samples the percentage of CD138⁺ myeloma cells is very low, adding more challenge for us. On the other hand, quantification of antigen expression in the form of molecules/cell is still based on the MFI value of antibody staining and therefore may be not so precise or really absolute as envisaged, and the value results may vary considerably in case different antibodies against the same antigen are used. Even when such a value of molecules/cell (eg. 10, 50 or 100) on normal cells is obtained, we still can not directly judge whether the expression level is really high or low and neither can we predict whether CAR-T cells can recognize the cells or not on the basis of the value. Moreover, illustration of the surface antigen expression levels in the form of relative or normalized MFI can be found in the papers published on the top-ranked Blood journal, as exemplified here: Blood (2011) 117 (3): 890–901; Blood. (2021) 138(19):1830-1842. Taken together, we would like to acknowledge that quantification of molecules/cells is a good means becoming more popular and meriting recommendation, yet it does not represent a must or a golden standard for determination of antigen expression by flow cytometry analysis. For the above-mentioned reasons, we would be very thankful if the reviewer could kindly understand our situation and thus allow us to acknowledge this as a limitation as indicated in Line 370-373 in the discussion part instead.

If there is no appropriate IHC FcRH5-specific antibody available to evaluate FcRH5 expression in human tissues: Could this data be extracted from the cancer genome atlas and the human protein atlas?

Response: Thanks a lot for the suggestion. TCGA generally does not provide the protein expression data to our knowledge. We have confirmed that the expression data of FcRH5 protein on human normal tissues is still not available from the human protein atlas (Pending normal tissue annotation as denoted). Although the FcRH5 protein expression data on human normal tissues remain unavailable, the expression pattern of FcRH5 transcript on human normal tissues has been reported previously. The early studies demonstrated that FcRH5 was preferentially expressed by B lineage cells in the spleen, tonsils, and lymph nodes by Northernblot analysis (Immunity. 2001 Mar;14(3):277-89.; Blood. 2002 Apr 15;99(8):2662-9.). It has been documented later that FcRH5 mRNA was selectively expressed in B lineage cells and tissues based on the analysis of GTEx sample set, consisting of 8555 samples from 544 donors over 53 tissues (Cancer Cell. 2017 Mar 13;31(3):383-395.). Extraction of the transcriptomics data of 54 human normal tissues from the HPA and GTEx datasets demonstrated that FcRH5 transcript was predominantly expressed in some lymphoid tissues including tonsil, lymph node, spleen and appendix, and was also detected at a much lower level in various gastrointestinal organs together with urinary bladder, salivary

gland, and thymus, while its expression in the remaining tissues or organs was quite low or undetectable, and this has been included in Supplementary Fig. 12a. Moreover, inspection of single cell RNA sequencing (scRNAseq) data from 29 human normal tissues and peripheral blood mononuclear cells consisting of 79 cell types revealed that analogous to CD19 and BCMA, FcRH5 mRNA was highly and exclusively expressed in plasma cells and B cells, as well as in the early spermatids to a lesser extent (Please see Supplementary Fig.12b). Therefore, we speculated that the mRNA signal detected in the above-mentioned tissues or organs was very likely derived from infiltrating B cells and plasma cells. Moreover, the data from the ongoing Phase I study (NCT03275103) of cevostamab, a FcRH5xCD3 bispecific antibody (BsAb) monotherapy in a large cohort of patients with heavily pre-treated RRMM demonstrated promising activity with durable response and manageable safety without causing serious damage to any essential organ or tissue, which may help considerably if not fully mitigate our concern about the safety of CAR-T cells targeting FcRH5. We have included the aforementioned statement in the discussion (Line 425-455).

Re: Data displayed for the FcRH5 expression in patient samples with and without 1q21 mutations (Supp. Fig. 1 a,b) : Here, as for the flow cytometric data in Fig. 1, a quantitative analysis of the expression levels (i.e. molecules/cells) is recommended.

Response: We appreciate the reviewer's suggestion, while we do not have the chance to perform this analysis for the same reason as described above.

Re: 2. In vitro analyses with FcRH5 CAR T cells:

Thanks for including the data for the CD4 vs. CD8 composition of the CAR-T cells and the clarification on the cytotoxicity assay. However, this information should be clearly stated in the main text and/or appropriate figure legends, especially as most researchers in the field would raise the same question regarding the cytolytic activity of these CAR-T cell products (could be also added in the supplement). Furthermore, a summary of the CD4 vs. CD8 composition from all donors used in this study should be included so that it is clear that CAR T cells used for all experiments had a similar CD4:CD8 ratio.

Response: Thanks a lot for the suggestion. We have incorporated the long-term cytotoxicity data of FcRH5 CAR-T cells against the patient-derived myeloma cells in the updated supplementary Figure 5a and added the statement that almost all the patient-derived primary myeloma cells were killed by autologous FcRH5 CAR-T cells following overnight co-culture, corroborating that the remaining target cells following short-term co-culture were less likely to represent "resistant" myeloma cells (Line 219-223). Meanwhile, we have stated in the main text that the CD4 vs. CD8 composition in between the mock T, mono-specific and bispecific CAR-T cells did not vary significantly for the donors used (Line 341-343), and also included a summary of the CD4 vs. CD8 composition from the donors used in this study in the updated supplementary Fig.10.

Re: 3. In vivo analyses with FcRH5 CAR T cells:

Thanks for including follow-up, BLI analyses etc. for the NCI-H929 s.c. myeloma in the revised manuscript. It is still highly recommended to use the same myeloma cell line for the disseminated

myeloma model. It should be possible to establish such a model also with the NCI-H929 cell line as exemplified e.g. here: Mirandola et al. BMC Cancer 2011, 11:394 and here: Proc Natl Acad Sci USA. 2019 Mar 5;116(10):4592-4598. doi: 10.1073/pnas.1821733116.

Response: Regarding the NCI-H929 disseminated MM model, we would like to thank the reviewer for raising two published articles to support that this model could be available. When going through the two articles, we noticed that in the BMC Cancer paper only the result of flow cytometry analysis was shown to support the existence of NCI-H929 cells in the circulation following i.v. injection; and in the PNAS paper, the authors were so lucky to catch the only one mouse with considerable tumor burden among four mice subject to i.v. injection, and then isolated the tumor cells from the bone marrow of the mouse for ex vivo expansion which were subsequently inoculated into a new batch of mice; and by this way, they were able to successfully establish the disseminated H929 MM model for evaluation of their drugs. While in our case, as we mentioned in the rebuttal letter previously, although we injected a large number of H929 cells up to 10M into 7 mice and waited for a couple of weeks, none of the mice developed apparent tumors as determined by BLI imaging, suggesting that we may have little chance to generate this model. Meanwhile, when checking the literature on Pubmed, we found that the PNAS paper may be the only one clearly demonstrating the successful establishment of NCI-H929 MM disseminated model. We have admitted in the updated manuscript that we failed to generate such a disseminated model with NCI-H929 cells and thus decided to use MM.1s-FcRH5 cells as an alternative (Line 251-256). We hope that the reviewer could kindly understand our situation.

Re: 4. Soluble FcRH5:

Thanks for the details, also for the bispecific CAR constructs. However, as others already established a sandwich ELISA to detect both isoforms, it would be highly recommended to inquire with the authors of this study if they could possibly share the antibodies and protocols to address this issue, especially for the detection of the isoforms in healthy vs. diseased states. This could also be interesting when using primary tumor material as targets instead of cell lines as this might also reveal which of the CAR constructs might be superior in a clinical setting.

Response: Thanks a lot for the comments and suggestion. Regarding the issue of soluble FcRH5, the current evidence strongly supports that IRTA2a may be very likely the predominant form of sFcRH5 as we already highlighted in the discussion part (Line 415-420). Meanwhile, we have mentioned that others previously reported measuring the concentrations of sFcRH5 in the MM patients v.s. normal donors with their home-made ELISA kit which is yet not available for us. On the other hand, please understand that we have actually tried our best to address the reviewer's concern by testing two commercially available ELISA kits although they were found to not work as expectedly. In this context, it may be not quite reasonable to still request us to set up the ELISA system by ourselves and then perform the similar experiment, which is apparently not the main focus of this manuscript. Alternatively, we have admitted this as a technical limitation in the discussion part (Line 397-401). In addition, we noticed that the reviewer just raised a new and interesting issue about using primary tumor material as targets instead of cell lines to help reveal which of the CAR constructs might be superior in a clinical setting, and would like to appreciate the reviewer's suggestion but consider testing it in a follow-up study.

Re: 5. Safety profile and requirement for a suicide gene

Thanks for the clarification why the suicide gene was added to the CAR construct. Re: expression data shown in Suppl. Fig. 13a: As already mentioned above: flow cytometric analysis without quantification is not sufficient to exclude possible low level expression of the target antigen on normal tissue. If mRNA data on expression of FcRH5 is already available, that should be referenced accordingly. Furthermore, as outlined above: if no suitable FcRH5 IHC antibody is available so far, please consider to check the cancer genome atlas and the human protein atlas if the data is available there. Re: cross-reactivity studies: Such a study should be performed on human tissue to demonstrate that the CAR construct does not recognize normal tissues.

Response: Thanks a lot for the valuable comments. We hope the reviewer can understand that we do not have the chance to recheck the molecules/cell due to lack of the original samples and really feel sorry about this. In addition, as we replied in the section of Re:1., quantification of molecules/cells is a good means becoming more popular and meriting recommendation, but it does not represent a must or a golden standard for determination of antigen expression by flow cytometry analysis. In fact, even when such an exact value of molecules/cell on normal cells is obtained, we still can not directly judge whether the expression level is really high or low and can not predict whether CAR-T cells can recognize the corresponding cells or not. On the other hand, as we mentioned above, presentation of antigen expression levels in the form of relative or normalized MFI is also widely applied, so we have incorporated the corresponding MFIR values in the Suppl. Fig. 13a for your reference.

The mRNA data on expression of FcRH5 in human normal tissues from others' report as well as the publically available datasets have been referenced accordingly in the text (Line 425-445). Regarding the suggestion to check the cancer genome atlas and the human protein atlas if no suitable FcRH5 IHC antibody is available so far, please see my detailed reply to Re:1. Regarding the cross-reactivity studies, although we have shown that FcRH5 CAR-T cells did not recognize normal donor-derived T cell, NK cells, monocytes as well as HPSCs by ELISA assay, we really can not perform such an assay to demonstrate that the CAR construct does not recognize various human normal tissues which are not available for us for ethical consideration.

REVIEWERS' COMMENTS

Reviewer #3 (expert in CAR T cells, bone marrow transplantation, hematological malignancies):

The authors have adequately addressed my comments and concerns.

RESPONSE TO REVIEWERS' COMMENTS

Reviewer #3 (expert in CAR T cells, bone marrow transplantation, hematological malignancies):

The authors have adequately addressed my comments and concerns.

Re: We appreciate the time and effort that the reviewer dedicated to providing valuable feedback on our manuscript and are grateful for the insightful comments and suggestion.